# Sampling Process Brings Additional Bias for Debiased Recommendation

## Abstract

In recommender systems, selection bias arises from the users' selective interactions with items, which poses a widely-recognized challenge for unbiased evaluation and learning for recommendation models. Recently, doubly robust and its variants have been widely studied to achieve debiased learning of prediction models. However, if the users and items in the training set are not exactly the same as those in the test set, even if the imputed errors and learned propensities are accurate, all previous doubly robust based debiasing methods are biased. To tackle this problem, in this paper, we first derive the bias of doubly robust learning methods and provide alternative unbiasedness conditions when users and items are sampled from a superpopulation. Then we propose a novel superpopulation doubly robust target learning approach (SuperDR), which is unbiased when either the imputation model or propensity model is correctly specified. We further derive the generalization error bound of the proposed method under superpopulation, and show that it can be effectively controlled by the proposed target learning approach. We conduct extensive experiments on three real-world datasets, including a large-scale industrial dataset, to demonstrate the effectiveness of our method.

## 1 Introduction

In the era of information explosion, recommender system (RS) plays an increasingly important role in areas such as e-commerce platforms, news reading, and social media. However, due to the subjective preferences of users and the data collection process itself, selection bias always exists in the collected data (Pradel et al., 2012), which poses a widely-recognized challenge (De Myttenaere et al., 2014; Marlin and Zemel, 2009). Ignoring selection bias makes RS difficult to provide accurate recommendations to users, thus hurting the user's experience and reducing social welfare.

Many methods have been proposed to address selection bias. The error imputation based (EIB) method (Chang et al., 2010; Marlin et al., 2007; Steck, 2010; 2013) utilizes an imputation model to impute the missing relevance. The inverse propensity score (IPS) method uses inverse propensity to reweight the observed events to achieve unbiasedness (Imbens and Rubin, 2015; Saito et al., 2020; Schnabel et al., 2016). The doubly robust (DR) method combines the error imputation model and the propensity model (Wang et al., 2019; Saito, 2020; Wang et al., 2022; Oosterhuis, 2023), which is unbiased if either the imputed errors or the learned propensities are accurate, which is also proved to has smaller variance compared to the IPS method (Saito, 2020; Oosterhuis, 2022).

Although previous methods have demonstrated promising performance in debiasing tasks, their unbiasedness relies on the assumption that the test set contains exactly the same users and items as the training set. As illustrated in Figure 1, if the users and items in the training set are randomly sampled from a larger superpopulation, previous debiasing methods are unbiased only if the test set contains exactly the same users and items, and are otherwise biased, even if the test set is another random sampling and the imputed errors and learned propensities are correct. For example, for the user side in the e-commerce platform, the training set contains users who are active and participate in transactions, while the whole population is all registered users. One may argue that the training set should contain all registered users, in this case, the whole population can be regarded as both the registered and non-registered users. In other words, we can always assume that there exists a larger population without loss of generality. At this point, for a set of users that have not appeared in the

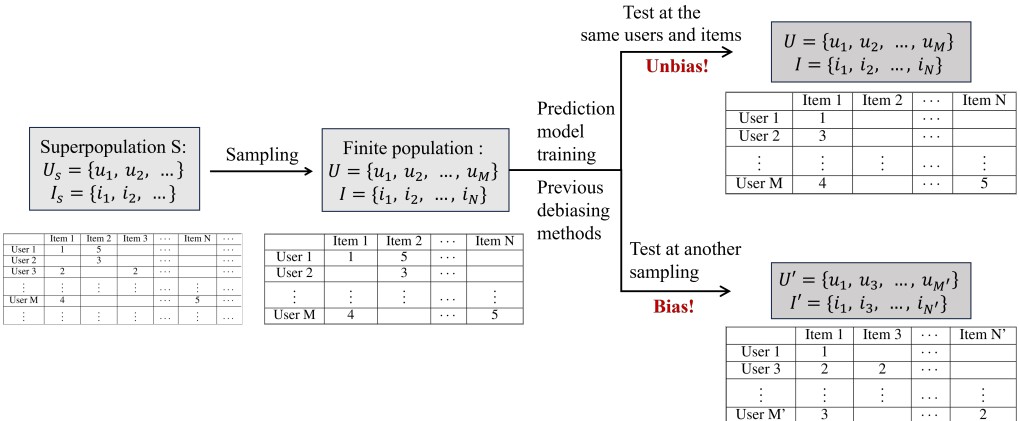

Figure 1: The drawbacks of previous debiasing methods.

training set, even if the distribution is consistent with the training set, previous debiasing methods still cannot achieve unbiased recommendations.

To this end, in this paper, we first derive the bias of doubly robust learning methods and provide alternative unbiasedness conditions when users and items are sampled from a superpopulation. Then we propose a novel superpopulation doubly robust (SuperDR) joint learning approach, which improves the accuracy of the imputed errors and leads to unbiased learning under probabilistic error imputations and learned propensities. We further derive the generalization error bound when using the probabilistic models, and show that it can be effectively controlled by the proposed learning approach. Extensive experiments are conducted on three real-world datasets, including a large-scale industrial dataset, to demonstrate the effectiveness of our proposal.

Our main contributions can be summarized as follows:

- To the best of our knowledge, this is the first paper that considers the randomness (thus the additional bias) introduced by the sampling process. In this scenario, we show the bias of the DR estimator has two terms: a covariance term and a term that measures the accuracy of learned propensities and imputed errors.
- In order to control the covariance term while obtaining accurately learned propensities and imputed errors, we propose the SuperDR method based on the target learning approach, which is unbiased under the new scenario and can effectively control the generalization bounds.
- We conduct extensive experiments on three real-world datasets, including a large industrial dataset, to demonstrate the effectiveness of our proposed method.

## 2 RELATED WORK

There are various biases in the data collected from RS (Chen et al., 2020; Wu et al., 2022), which have been of increasing concern in recent years (Ai et al., 2018; Saito and Nomura, 2022; Liu et al., 2021; Zhang et al., 2021; Luo et al., 2021; Liu et al., 2022; Lin et al., 2023). Selection bias is one of the most common biases in RS and a lot of research has been done aiming to eliminate this kind of bias (Chen et al., 2021; Guo et al., 2021; Liu et al., 2020; Saito, 2020; Schnabel et al., 2016; Wang et al., 2019). The error imputation based method (EIB) (Chang et al., 2010; Marlin et al., 2007; Steck, 2010; 2013) first imputes pseudo-labels for missing events from the observed events, and then leverages these pseudo-labels to train the prediction model (Dudík et al., 2011; Marlin et al., 2007; Steck, 2013; Wu et al., 2022). An alternative way to eliminate selection bias is to weight the inverse propensity score (IPS) on the observed data to eliminate bias (Imbens and Rubin, 2015; Saito et al., 2020; Schnabel et al., 2016). However, IPS will suffer from a large variance when the extreme values exist in the estimated propensities (Thomas and Brunskill, 2016).

The doubly robust (DR) method improves the weakness of EIB and IPS methods and becomes the mainstream model due to the weaker unbiasedness conditions and smaller variance than the IPS

method (Benkeser et al., 2017; Morgan and Winship, 2015; Luo et al., 2021; Li et al., 2023d; Saito, 2020; Wang et al., 2019). In particular, the DR estimator is unbiased when either learned propensities or imputed errors are accurate. Many augmented DR methods are developed to further enhance the previous DR method performance by modifying the propensity model and imputation model or the form of the DR estimator, such as MRDR (Guo et al., 2021), BRD-DR (Ding et al., 2022), StableDR (Li et al., 2023d), TDR (Li et al., 2023b), DR-MSE (Dai et al., 2022), and DR-BIAS (Dai et al., 2022). However, these approaches are limited to the use of deterministic error imputation and propensity models and fail to be unbiased when using probabilistic models to impute errors and learn propensities. To the best of our knowledge, this is the first paper that extends previous widely adopted DR methods to be compatible with probabilistic error imputation and propensity models.

## 3 PRELIMINARIES

We start with the classic scenario. Suppose the user set $\mathcal{U} = \{u_1, u_2, \ldots, u_m\}$ contains $m$ users, the item set $\mathcal{I} = \{i_1, i_2, \ldots, i_n\}$ contains $n$ items, and denote the set of all user-item pairs as $\mathcal{D} = \mathcal{U} \times \mathcal{I}$. Let $\mathbf{R} \in \mathbb{R}^{m \times n}$ be the ground truth rating matrix of all user-item pairs, where $r_{u,i}$ is the rating of user $u$ on item $i$. Let $x_{u,i}$ be the feature of user $u$ and item $i$, and $\hat{r}_{u,i} = f(x_{u,i}; \theta)$ is the predicted rating by a prediction model, $\theta$ is the corresponding parameter. Denote $\hat{\mathbf{R}} \in \mathbb{R}^{m \times n}$ as the matrix contains all the predicted ratings. Let $\mathbf{O} \in \{0, 1\}^{m \times n}$ be the binary observation indicator matrix for all user-item pairs, $o_{u,i} = 1$ indicates the rating of user $u$ on item $i$ is observed, otherwise missing $o_{u,i} = 0$. The purpose of RS is to train a prediction model to accurately predict all ratings. If all the ratings are observed, the prediction model can be trained directly by minimizing the ideal loss

$$\mathcal{L}_{ideal}(\theta) = \frac{1}{|\mathcal{D}|} \sum_{(u,i) \in \mathcal{D}} e_{u,i},$$

where $e_{u,i} = \mathcal{L}(\hat{r}_{u,i}, r_{u,i})$ is the loss between the predicted rating $\hat{r}_{u,i}$ and the true rating $r_{u,i}$ and $\mathcal{L}(\cdot, \cdot)$ is an arbitrary loss function. However, the ideal loss is not available in most cases because we can only observe partial biased data. To tackle this issue, the DR estimator has been proposed:

$$\mathcal{E}_{\mathrm{DR}}(\theta) = \frac{1}{|\mathcal{D}|} \sum_{(u,i) \in \mathcal{D}} \left[ \hat{e}_{u,i} + \frac{o_{u,i}(e_{u,i} - \hat{e}_{u,i})}{\hat{p}_{u,i}} \right].$$

where $\hat{p}_{u,i} = \pi(x_{u,i}; \psi)$ is the propensity model to estimate $p_{u,i} := \mathbb{P}(o_{u,i} = 1 \mid x_{u,i})$, and $\hat{e}_{u,i}$ is the imputation model to impute the missing $e_{u,i}$.

Below, we focus on the theoretical properties of the DR estimator and start from the widely-known conclusions for the bias form for DR estimator.

**Lemma 1** (Bias of DR Estimator (Wang et al., 2019)). *Given imputed errors $\hat{e}_{u,i}$ and learned propensities $\hat{p}_{u,i} > 0$ for all user-item pairs, when considering only the randomness of rating missing indicators, the bias of the DR estimator is*

$$\mathrm{Bias}_{\mathbf{O}}[\mathcal{E}_{\mathrm{DR}}(\theta)] = \frac{1}{|\mathcal{D}|} \sum_{(u,i) \in \mathcal{D}} \frac{\{\hat{p}_{u,i} - p_{u,i}\} \cdot \{e_{u,i} - \hat{e}_{u,i}\}}{\hat{p}_{u,i}}.$$

We find that either $\hat{e}_{u,i} = e_{u,i}$ or $\hat{p}_{u,i} = p_{u,i}$ is sufficient to eliminate bias under deterministic models, which inspires the double robustness condition for the DR method.

**Corollary 1** (Double Robustness (Wang et al., 2019)). *The DR estimator is unbiased when either imputed errors $\hat{e}_{u,i}$ or learned propensities $\hat{p}_{u,i}$ are accurate, i.e., either $\hat{e}_{u,i} = e_{u,i}$ or $\hat{p}_{u,i} = p_{u,i}$.*

## 4 PROPOSED METHOD

### 4.1 FROM FINITE POPULATION TO SUPERPOPULATION

The above Lemma 1 shows the bias form of the DR estimator when users and items in the training set and the users and items in the test set are exactly the same. However, as we discussed

earlier, this assumption does not hold in many real-world scenarios. Formally, in such a general scenario, we denote $\mathcal{U} = \{u_1, u_2, ...\}$, $\mathcal{I} = \{i_1, i_2, ...\}$ are the user set and item set, and $\mathcal{D}_{\text{train}} = \{u_1, u_2, \ldots, u_m\} \times \{i_1, i_2, \ldots, i_n\}$ and $\mathcal{D}_{\text{test}} = \{u_{j_1}, u_{j_1}, \ldots, u_{j_{m'}}\} \times \{i_{k_1}, i_{k_2}, \ldots, i_{k_{n'}}\}$ are sampled from the whole user set and item set, respectively. Without loss of generality, we assume that the sampling strategy is the same for both $D_{\text{train}}$ and $D_{\text{test}}$ datasets (otherwise, we can adjust the sampling strategy by using reweighting). Note that the learned imputed error $\hat{e}_{u,i}$ no longer estimates $e_{u,i}$, but estimates the error expectation $\mathbb{E}(e_{u,i} \mid x_{u,i})$, and the learned propensity $\hat{p}_{u,i}$ estimates $\mathbb{E}(p_{u,i} \mid x_{u,i})$. The following theorem and corollary show the bias and the adjusted DR property for the DR estimator.

**Theorem 1** (Bias of DR Estimator under **Superpopulation**). *Given probabilistic error imputation model $\hat{e}_{u,i}$ and probabilistic propensity model $\hat{p}_{u,i}$, consider all variables are random, then the bias of the DR estimator is*

$$\text{Bias}_{\mathcal{P}}[\mathcal{E}_{\text{DR}}(\theta)] = \underbrace{\text{Cov}\left(\frac{\hat{p}_{u,i} - o_{u,i}}{\hat{p}_{u,i}}, e_{u,i} - \hat{e}_{u,i}\right)}_{\text{equals to 0 if independent}}$$

$$+ \underbrace{\mathbb{E}\left[\left\{1 - \mathbb{E}\left[\frac{o_{u,i}}{\hat{p}_{u,i}}\Big|x_{u,i}\right]\right\} \cdot \{\mathbb{E}[e_{u,i} \mid x_{u,i}] - \mathbb{E}[\hat{e}_{u,i} \mid x_{u,i}]\}\right]}_{\text{equals to 0 either } \mathbb{E}[o_{u,i}/\hat{p}_{u,i} \mid x_{u,i}] = 1 \text{ or } \mathbb{E}[\hat{e}_{u,i} - e_{u,i} \mid x_{u,i}] = 0}$$

**Corollary 2** (Double Robustness under **Superpopulation**). *The DR estimator is unbiased when both the following conditions hold:*

*(i) The covariance term vanishes,* i.e., $\text{Cov}\left(\frac{\hat{p}_{u,i} - o_{u,i}}{\hat{p}_{u,i}}, e_{u,i} - \hat{e}_{u,i}\right) = 0$;

*(ii) Either learned propensities satisfy $\mathbb{E}[o_{u,i}/\hat{p}_{u,i} \mid x_{u,i}] = 1$, or imputed errors have the same conditional expectation with true prediction errors $\mathbb{E}[\hat{e}_{u,i} \mid x_{u,i}] = \mathbb{E}[e_{u,i} \mid x_{u,i}]$.*

Compared with the existing theoretical results as in Lemma 1, it is obvious that condition *(ii)* is necessary to achieve unbiasedness, which directly extends the conditions of accurate imputed errors and learned propensities in Lemma 1 to the expectation form. However, note that the condition *(i)* that covariance vanishes is also needed for the unbiasedness under superpopulation scenario. Therefore, it is necessary to modify the previous DR learning approach to control the covariance and simultaneously learn accurate propensity and imputation models.

## 4.2 THE PROBABILISTIC DR ESTIMATOR

It is important to note that the true covariance is unknown because we cannot access the true data distribution. However, we can use the empirical covariance over all user-item pairs as an approximation of the true covariance. We first give the definition of empirical covariance.

**Definition 1** (Empirical Covariance). *The empirical expected conditional covariance between $(\hat{p}_{u,i} - o_{u,i})/\hat{p}_{u,i}$ and $e_{u,i} - \hat{e}_{u,i}$ is*

$$\widehat{\text{Cov}}\left(\frac{\hat{p}_{u,i} - o_{u,i}}{\hat{p}_{u,i}}, e_{u,i} - \hat{e}_{u,i}\right) = \frac{1}{|\mathcal{D}|} \sum_{(u,i)\in\mathcal{D}} \frac{\hat{p}_{u,i} - o_{u,i}}{\hat{p}_{u,i}} \cdot (e_{u,i} - \hat{e}_{u,i}).$$

A direct method to control the empirical covariance is to regard it as a regularization term. However, since the data are partially observed, we cannot obtain the value of the empirical covariance on all user-item pairs. In addition, the large penalty term may hurt the prediction performance. Interestingly, we found that the empirical covariance can be controlled with subtle changes to the DR estimator. Specifically, we designed imputation balancing correction as follows:

$$\tilde{e}_{u,i} = m(x_{u,i}; \phi) + \epsilon(o_{u,i} - \pi(x_{u,i}; \psi)).$$

Motivated by targeted maximum likelihood estimation (van der Laan and Rose, 2011), we add a correction term $\epsilon(o_{u,i} - \pi(x_{u,i}; \psi))$ on $\hat{e}_{u,i}$, which has zero mean under accurate $\pi(x_{u,i}; \psi)$, thus will not bring extra bias to the imputation model. We then learn $\phi$ and $\epsilon$ in $\tilde{e}_{u,i}$ by minimizing

$$(\phi^*, \epsilon^*) = \arg\min_{\phi,\epsilon} \mathcal{L}_e^{Bal}(\phi, \epsilon) = \frac{1}{|\mathcal{D}|} \sum_{(u,i)\in\mathcal{D}} \frac{o_{u,i}(e_{u,i} - \tilde{e}_{u,i})^2}{\hat{p}_{u,i}} + v\|\phi\|_F^2,$$

where $\| \cdot \|_F^2$ is the Frobenius norm. This proposed loss has several desired properties. First, the derivatives on the proposed loss with respect to $\epsilon$ are shown below:

$$\frac{\partial}{\partial \epsilon} \mathcal{L}_e^{Bal}(\phi, \epsilon) = \frac{2}{|\mathcal{D}|} \sum_{(u,i) \in \mathcal{O}} \frac{\hat{p}_{u,i} - o_{u,i}}{\hat{p}_{u,i}} \cdot (e_{u,i} - \tilde{e}_{u,i}).$$

It has the same form as the empirical covariance for user-item pairs with $o_{u,i} = 1$, which means that we can make the empirical covariance for observed user-item pairs to exact zero by minimizing the $\mathcal{L}_e^{Bal}$ directly. Meanwhile, the gradient contains $\epsilon$ when taking the derivatives with respect to $\phi$, which indicates a well-learned $\epsilon$ can lead to a more accurate $\phi$ to further ensure unbiasedness. Moreover, the unobserved empirical covariance can also be bounded by $\mathcal{L}_e^{Bal}$ using the concentration inequality. Theorem 2 below shows the controllability of empirical covariance.

**Theorem 2** (Controllability of Empirical Covariance). *The boosted imputation model trained by the balanced enhanced imputation loss is sufficient for controlling the empirical covariance.*

*(i) For user-item pairs with* **observed** *outcomes, the empirical covariance is 0. Formally, we have*

$$\frac{\partial}{\partial \epsilon} \mathcal{L}_e^{Bal}(\phi, \epsilon) \bigg|_{\epsilon = \epsilon^*} = 0, \text{ which is equivalent to } \frac{1}{|\mathcal{D}|} \sum_{(u,i): o_{u,i}=1} \frac{\hat{p}_{u,i} - o_{u,i}}{\hat{p}_{u,i}} \cdot (e_{u,i} - \tilde{e}_{u,i}) = 0;$$

*(ii) For user-item pairs with* **missing** *outcomes, suppose that $\hat{p}_{u,i} \geq K_\psi$ and $|e_{u,i} - \tilde{e}_{u,i}| \leq K_\phi$, then with probability at least $1 - \eta$, we have*

$$\frac{1}{|\mathcal{D}|} \sum_{(u,i): o_{u,i}=0} \frac{\hat{p}_{u,i} - o_{u,i}}{\hat{p}_{u,i}} \cdot (e_{u,i} - \tilde{e}_{u,i}) \leq \sqrt{\mathcal{L}_e^{Bal}(\phi, \epsilon)} + K_\phi \cdot \sqrt{\frac{1}{|\mathcal{D}|} \sum_{u,i \in \mathcal{D}} \left| 1 - \mathbb{E}\left[ \frac{o_{u,i}}{\hat{p}_{u,i}} \Big| x_{u,i} \right] \right|}$$

$$+ \sqrt{K_\phi \left( 1 + \frac{1}{K_\psi} \right) \left( 2\mathcal{R}(\mathcal{F}) + (2K_\phi + 1) \sqrt{\frac{2 \log(4/\eta)}{|\mathcal{D}|}} \right)}.$$

Note that the proposed imputation balancing correction has no harm property. That is, when the $\hat{e}_{u,i}$ has already ensured the empirical covariance to zero, the $\epsilon$ will converge to zero to degrade.

**Corollary 3** (Relation to previous imputed errors). *The learned coefficient $\epsilon^*$ will converge to zero when the probabilistic imputation model $\hat{e}_{u,i}$ has already led to zero empirical covariance, making $\tilde{e}_{u,i}$ degenerates to $\hat{e}_{u,i}$.*

In addition, Corollary 4 shows that the proposed imputation balancing correction can not only control the empirical covariance effectively but also be helpful for learning more accurate imputed errors when the previous imputed errors are inaccurate.

**Corollary 4** (Bias reduction property). *The proposed balancing enhanced imputation loss leads to the smaller bias of imputed errors $\tilde{e}_{u,i}$, when $\hat{e}_{u,i}$ are inaccurate. Formally, we have*

$$\min_{\phi, \epsilon} \mathcal{L}_e^{Bal}(\phi, \epsilon) = \frac{1}{|\mathcal{D}|} \sum_{(u,i) \in \mathcal{D}} \frac{o_{u,i}(e_{u,i} - \tilde{e}_{u,i})^2}{\hat{p}_{u,i}} \leq \min_{\phi} \mathcal{L}_e(\phi) = \frac{1}{|\mathcal{D}|} \sum_{(u,i) \in \mathcal{D}} \frac{o_{u,i}(e_{u,i} - \hat{e}_{u,i})^2}{\hat{p}_{u,i}}.$$

Moreover, while reducing bias, the proposed method also reduces the variance compared to the previous imputed errors under a moderate condition, as shown below.

**Corollary 5** (Variance reduction property). *The proposed balancing enhanced imputation loss leads to the smaller variance of $\tilde{e}_{u,i}$ when the optimal $\epsilon^*$ lies in a certain range. Formally, we have*

$$\mathbb{V}(\tilde{e}_{u,i}) = \mathbb{V}(\hat{e}_{u,i} + \epsilon^* \cdot (o_{u,i} - \hat{p}_{u,i})) \leq \mathbb{V}(\hat{e}_{u,i}), \text{if } \epsilon^* \in \left[ 0, 2 \cdot \frac{\text{Cov}(\hat{e}_{u,i}, \hat{p}_{u,i} - o_{u,i})}{\mathbb{V}(\hat{p}_{u,i} - o_{u,i})} \right].$$

Finally, the proposed SuperDR estimator is given as

$$\mathcal{E}_{\text{SuperDR}}(\theta) = \frac{1}{|\mathcal{D}|} \sum_{(u,i) \in \mathcal{D}} \left[ \tilde{e}_{u,i} + \frac{o_{u,i}(e_{u,i} - \tilde{e}_{u,i})}{\hat{p}_{u,i}} \right],$$

where $\tilde{e}_{u,i} = m(x_{u,i}; \phi) + \epsilon(o_{u,i} - \pi(x_{u,i}; \psi))$.

---

**Algorithm 1:** The Proposed **Superpopulation** Doubly Robust Joint Learning

---

**Input:** observed ratings $\mathbf{R}^o$ and a pre-trained probabilistic propensity model $\pi(x_{u,i}; \psi)$.

1 **while** *stopping criteria is not satisfied* **do**
2     **for** *number of steps for training the balancing enhanced imputation model* **do**
3         Sample a batch of user-item pairs $\{(u_j, i_j)\}_{j=1}^J$ from $\mathcal{O}$;
4         Update $\phi$ by descending along the gradient $\nabla_\phi \mathcal{L}_e^{Bal}(\phi, \epsilon)$;
5         **Update $\epsilon$ by descending along the gradient $\nabla_\epsilon \mathcal{L}_e^{Bal}(\phi, \epsilon)$;**
6     **end**
7     **for** *number of steps for training the debiased prediction model* **do**
8         Sample a batch of user-item pairs $\{(u_k, i_k)\}_{k=1}^K$ from $\mathcal{D}$;
9         Update $\theta$ by descending along the gradient $\nabla_\theta \mathcal{L}_{\text{SuperDR}}(\theta; \phi, \psi)$;
10     **end**
11 **end**

---

## 4.3 THE LEARNING ALGORITHM

We optimize the prediction model and the imputation model of the SuperDR method by a widely used joint learning framework (Wang et al., 2019), which alternatively optimizes two models to achieve unbiased learning. Specifically, we train prediction model by minimizing the SuperDR loss:

$$\mathcal{L}_{\text{SuperDR}}(\theta) = \frac{1}{|\mathcal{D}|} \sum_{(u,i) \in \mathcal{D}} \left[ \tilde{e}_{u,i} + \frac{o_{u,i}(e_{u,i} - \tilde{e}_{u,i})}{\hat{p}_{u,i}} \right] + v\|\theta\|_F^2.$$

We update the imputation model parameters and $\epsilon$ simultaneously by minimizing the $\mathcal{L}_e^{Bal}(\phi, \epsilon)$ in Section 4.2. The parameters of the prediction and imputation model are updated alternatively via stochastic gradient descent. The joint learning process is summarized in Algorithm 1.

## 4.4 THE GENERALIZATION BOUND

Next, we analyze the generalization error bound of the DR methods using the probabilistic models for estimating $e_{u,i}$ and $p_{u,i}$, and show that controlling empirical covariance leads to a tighter bound. Specifically, the generalization error theories for the previous DR estimators relied mainly on the boundedness of the loss to each user-item pair in the DR estimators from the binary indicator $o_{u,i}$, *i.e.*, for the DR estimator, the bound for DR loss on $(u, i)$ is $(e_{u,i} - \hat{e}_{u,i})/\hat{p}_{u,i}$. However, these analyses no longer hold under superpopulation scenario. To proceed, we first define the empirical Rademacher complexity as below.

**Definition 2** (Empirical Rademacher Complexity (Shalev-Shwartz and Ben-David, 2014)). *Let $\mathcal{F}$ be a family of prediction models mapping from $x \in \mathcal{X}$ to $[a, b]$, and $S = \{x_{u,i} \mid (u, i) \in \mathcal{D}\}$ a fixed sample of size $|\mathcal{D}|$ with elements in $\mathcal{X}$. Then, the empirical Rademacher complexity of $\mathcal{F}$ with respect to the sample $S$ is defined as:*

$$\mathcal{R}(\mathcal{F}) = \mathbb{E}_{\boldsymbol{\sigma} \sim \{-1,+1\}^{|\mathcal{D}|}} \sup_{f_\theta \in \mathcal{F}} \left[ \frac{1}{|\mathcal{D}|} \sum_{(u,i) \in \mathcal{D}} \sigma_{u,i} e_{u,i} \right],$$

*where $\boldsymbol{\sigma} = \{\sigma_{u,i} : (u, i) \in \mathcal{D}\}$, and $\sigma_{u,i}$ are independent uniform random variables taking values in $\{-1, +1\}$. The random variables $\sigma_{u,i}$ are called Rademacher variables.*

Finally, we provide the generalization error bound of SuperDR, which includes four terms: the SuperDR loss itself, the empirical covariance, the bias of the SuperDR estimator, and the tail bound. Compared to the previous DR method, the proposed method can further control the covariance term, which leads to a more desirable generalization bound and thus improving the debiasing performance.

Table 1: Performance on AUC, NDCG@K and Recall@K on the **Coat**, **Yahoo** and **KuaiRec** datasets. The best result is bolded and the best baseline result is underlined, where * means statistically significant results (p-value $\leq 0.05$) using the paired-t-test.

| Methods | Coat | | | Yahoo | | | KuaiRec | | |
|---|---|---|---|---|---|---|---|---|---|
| | AUC | NDCG@5 | Recall@5 | AUC | NDCG@5 | Recall@5 | AUC | NDCG@50 | Recall@50 |
| Base | $0.718 \pm 0.003$ | $0.639 \pm 0.015$ | $0.612 \pm 0.010$ | $0.664 \pm 0.002$ | $0.645 \pm 0.002$ | $0.442 \pm 0.004$ | $0.808 \pm 0.005$ | $0.610 \pm 0.007$ | $0.645 \pm 0.010$ |
| DAMF | $0.722 \pm 0.008$ | $0.640 \pm 0.010$ | $0.617 \pm 0.007$ | $0.664 \pm 0.002$ | $0.642 \pm 0.001$ | $0.438 \pm 0.002$ | $0.811 \pm 0.003$ | $0.609 \pm 0.004$ | $0.643 \pm 0.005$ |
| CVIB | $0.725 \pm 0.007$ | $0.644 \pm 0.010$ | $0.620 \pm 0.007$ | $0.670 \pm 0.004$ | $0.656 \pm 0.003$ | $0.452 \pm 0.001$ | $0.816 \pm 0.007$ | $0.617 \pm 0.008$ | $0.653 \pm 0.009$ |
| IPS | $0.716 \pm 0.007$ | $0.640 \pm 0.006$ | $0.613 \pm 0.008$ | $0.667 \pm 0.003$ | $0.647 \pm 0.006$ | $0.445 \pm 0.007$ | $0.806 \pm 0.006$ | $0.606 \pm 0.006$ | $0.643 \pm 0.005$ |
| SNIPS | $0.713 \pm 0.003$ | $0.639 \pm 0.009$ | $0.613 \pm 0.010$ | $0.665 \pm 0.003$ | $0.644 \pm 0.004$ | $0.443 \pm 0.003$ | $0.811 \pm 0.004$ | $0.612 \pm 0.006$ | $0.649 \pm 0.006$ |
| ASIPS | $0.720 \pm 0.008$ | $0.639 \pm 0.004$ | $0.619 \pm 0.007$ | $0.668 \pm 0.002$ | $0.655 \pm 0.004$ | $0.452 \pm 0.005$ | $0.811 \pm 0.006$ | $0.614 \pm 0.006$ | $0.652 \pm 0.005$ |
| IPS-V2 | $0.717 \pm 0.004$ | $0.643 \pm 0.010$ | $0.622 \pm 0.007$ | $0.662 \pm 0.003$ | $0.651 \pm 0.001$ | $0.445 \pm 0.002$ | $0.813 \pm 0.006$ | $0.612 \pm 0.008$ | $0.655 \pm 0.006$ |
| DR | $0.721 \pm 0.004$ | $0.645 \pm 0.007$ | $0.621 \pm 0.007$ | $0.667 \pm 0.005$ | $0.655 \pm 0.004$ | $0.449 \pm 0.008$ | $0.818 \pm 0.003$ | $0.620 \pm 0.004$ | $0.655 \pm 0.007$ |
| MRDR | $0.720 \pm 0.006$ | $0.646 \pm 0.006$ | $\underline{0.624 \pm 0.007}$ | $0.665 \pm 0.005$ | $0.652 \pm 0.005$ | $0.448 \pm 0.005$ | $0.814 \pm 0.006$ | $0.616 \pm 0.008$ | $0.652 \pm 0.003$ |
| DR-MSE | $0.720 \pm 0.001$ | $0.639 \pm 0.008$ | $0.621 \pm 0.009$ | $0.667 \pm 0.004$ | $0.650 \pm 0.004$ | $0.446 \pm 0.004$ | $0.814 \pm 0.006$ | $0.617 \pm 0.006$ | $0.654 \pm 0.007$ |
| DR-V2 | $\underline{0.726 \pm 0.007}$ | $0.646 \pm 0.010$ | $0.621 \pm 0.009$ | $0.671 \pm 0.008$ | $\underline{0.660 \pm 0.005}$ | $\underline{0.456 \pm 0.003}$ | $0.821 \pm 0.010$ | $0.619 \pm 0.010$ | $\underline{0.661 \pm 0.008}$ |
| SDR | $0.722 \pm 0.005$ | $0.644 \pm 0.005$ | $0.623 \pm 0.010$ | $0.666 \pm 0.005$ | $0.653 \pm 0.004$ | $0.451 \pm 0.004$ | $0.819 \pm 0.004$ | $0.618 \pm 0.005$ | $0.652 \pm 0.006$ |
| TDR | $0.724 \pm 0.005$ | $0.643 \pm 0.006$ | $0.623 \pm 0.009$ | $0.664 \pm 0.004$ | $0.655 \pm 0.007$ | $0.453 \pm 0.003$ | $0.822 \pm 0.005$ | $0.621 \pm 0.009$ | $0.656 \pm 0.010$ |
| MR | $0.725 \pm 0.007$ | $\underline{0.647 \pm 0.006}$ | $0.622 \pm 0.007$ | $\underline{0.672 \pm 0.003}$ | $0.657 \pm 0.003$ | $0.454 \pm 0.002$ | $\underline{0.823 \pm 0.003}$ | $\underline{0.622 \pm 0.004}$ | $0.655 \pm 0.005$ |
| SuperDR | $\mathbf{0.739^* \pm 0.004}$ | $\mathbf{0.654^* \pm 0.005}$ | $\mathbf{0.626 \pm 0.010}$ | $\mathbf{0.673 \pm 0.003}$ | $\mathbf{0.662 \pm 0.003}$ | $\mathbf{0.459^* \pm 0.003}$ | $\mathbf{0.824 \pm 0.006}$ | $\mathbf{0.631^* \pm 0.005}$ | $\mathbf{0.679^* \pm 0.010}$ |

**Theorem 3** (Generalization Bound under **Superpopulation**). *Suppose that $\hat{p}_{u,i} \geq K_\psi$ and $|e_{u,i} - \hat{e}_{u,i}| \leq K_\phi$, then with probability at least $1 - \eta$, we have*

$$\mathcal{L}_{ideal}(\theta) \leq \mathcal{L}_{SuperDR}(\theta) + \widehat{\mathrm{Cov}}\left(\frac{\hat{p}_{u,i} - o_{u,i}}{\hat{p}_{u,i}}, e_{u,i} - \hat{e}_{u,i}\right) +$$

$$\mathrm{Bias}_{\mathcal{P}}(\mathcal{E}_{SuperDR}(\theta)) + \left(1 + \frac{1}{K_\psi}\right)\left(2\mathcal{R}(\mathcal{F}) + K_\phi\sqrt{\frac{18}{|\mathcal{D}|}\log\frac{4}{\eta}}\right).$$

## 5 EXPERIMENTS

### 5.1 EXPERIMENTAL SETUP

**Dataset and Preprocessing.** To verify the effectiveness of the proposed method in the real-world dataset, the dataset that contains both biased and unbiased data is required. Following the previous studies (Saito, 2020; Wang et al., 2019; 2021; Chen et al., 2021), the following three widely used real-world datasets are adopted to conduct our experiments: **Coat** [1] contains ratings from 290 users to 300 items. Each user rates 24 of the coats that are selected by themselves, which produces 6,960 biased ratings in total. Meanwhile, each user is asked to rate 16 randomly picked items, which generates 4,640 unbiased ratings. **Yahoo**[2] contains ratings from 15,400 users to 1,000 items. Each user rates several items to generate the 311,704 biased ratings. In addition, the first 5,400 users are asked to rate 10 randomly picked items, which constitutes the 54,000 unbiased ratings. We binarize the ratings to 0 for ratings less than 3, otherwise 1. We further use a fully exposed industrial dataset **KuaiRec**[3] (Gao et al., 2022) with 4,676,570 video watching ratio records from 1,411 users to 3,327 videos. For this dataset, we binarize the records to 0 for records less than 2, otherwise 1.

**Baselines.** In our experiments, we first use the matrix factorization (MF) (Mnih and Salakhutdinov, 2007) to generate the embedding for each user and item, and then fix such embedding as the user-item feature. Then we take the MLP for the base model and compared the proposed method with the following baselines **DAMF** (Saito and Nomura, 2022), the information bottleneck based method: **CVIB** (Wang et al., 2020), the propensity based methods: **IPS** (Schnabel et al., 2016), **SNIPS** (Swaminathan and Joachims, 2015), **ASIPS** (Saito, 2020), and **IPS-V2** (Li et al., 2023c), and the DR-based methods: **DR** (Wang et al., 2019), **MRDR** (Guo et al., 2021), **DR-MSE** (Dai et al., 2022), **DR-V2** (Li et al., 2023c), **TDR** (Li et al., 2023b), **SDR** (Li et al., 2023d), and **MR** (Li et al., 2023a).

**Experimental Protocols and Details.** The following three metrics are used to measure the debiasing performance: AUC, NDCG@K, and Recall@K, where we set K = 5 for **Coat** and **Ya-**

---

[1] https://www.cs.cornell.edu/˜schnabts/mnar/

[2] http://webscope.sandbox.Music.com/

[3] https://github.com/chongminggao/KuaiRec

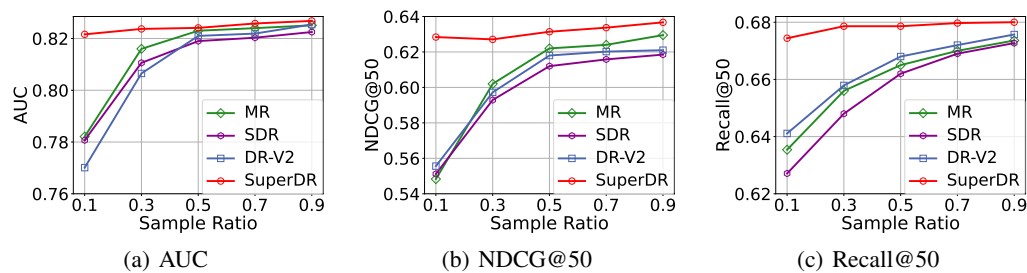

(a) AUC          (b) NDCG@50          (c) Recall@50

Figure 2: Effects of varying sample ratios on performance on the **KuaiRec** dataset.

**hoo**, while set K = 50 for **KuaiRec**. All the experiments are implemented on PyTorch with the GeForce RTX 3090 as the computational resource. Adam is utilized as the optimizer for fast convergence in all experiments. To simulate the superpopulation scenario, we first randomly sample $b\%$ users and items ($b$ is set to $50\%$ in our experiments except in Figures 2 and 3.) from the training set and then use the whole test set to evaluate the debiasing performance. Note that this intervention will not affect the data sparsity, it will only affect the number of observed users and items. In addition, we tune learning rate in $\{0.001, 0.005, 0.01, 0.05, 0.1\}$, batch size in $\{128, 256, 512\}$ for **Coat** and $\{1024, 2048, 4096\}$ for **Yahoo** and **KuaiRec**. The weight decay is tuned in $\{1e-5, 5e-5, \ldots, 1e-2\}$. In addition, We use the logistic regression model as the propensity model, which means that there is no unbiased data requirement for our method. [4]

### 5.2 PERFORMANCE COMPARISON

Table 1 summarizes the debiasing performance of various methods on three benchmark datasets **Coat**, **Yahoo**, and **KuaiRec**, and we have the following findings. First, most debiased methods outperform the base model, which shows the necessity for debiasing. Second, overall speaking, the information bottleneck-based methods perform slightly better than the propensity-based methods, while DR-based methods such as DR-V2 and MR demonstrate the most competitive performance, indicating the superiority of DR methods over other baselines. Third, the proposed SuperDR method achieves the best performance in terms of all evaluation metrics. This indicates that the SuperDR method can effectively reduce the additional bias introduced by sampling through controlling empirical covariance, and achieve an unbiased estimate of the ideal loss in scenarios where users and items in the training set are not exactly the same as those in the test set.

### 5.3 IN-DEPTH ANALYSIS

**Effects of Varying Bias Level.** Figures 2 investigates the impact of different levels of bias introduced by sampling on prediction performance on the **KuaiRec** dataset. We change the sample ratios to control the degree of overlap between users and items in the training and test sets. A higher sample ratio indicates a greater proportion of the same users and items in both sets, resulting in less bias introduced by sampling. When the sample ratio is 1, it means that the users and items in the training and test sets are identical, with no bias introduced by sampling. At this point, our method slightly outperforms recently proposed state-of-the-art methods such as DR-V2. When the sample ratio is 0.1 and 0.3, there are few overlapping users and items between the training and test sets, resulting in significant bias introduced by sampling. The performance of previous methods noticeably declines, while the SuperDR method effectively addresses this bias, achieving significant performance improvements. See more experiment results on **Yahoo** dataset in Appendix B.

**Effects of Empirical Covariance Control.** We explore the effects of Empirical Covariance (EC) Reduction on the prediction performance in Figure 3. We find that SuperDR achieves the most significant empirical covariance decreases and the most competitive performance in AUC and NDCG@K, which empirically demonstrates that the EC reduction contributes to the prediction performance. Note that TDR method obtains some performance improvement compared to vallina DR, this is because it adds $o_{u,i}(\frac{1}{\hat{p}_{u,i}} - 1)$ as the correction term to the imputed errors to control the co-

---

[4]Code will be fully open sourced once the paper is accepted.

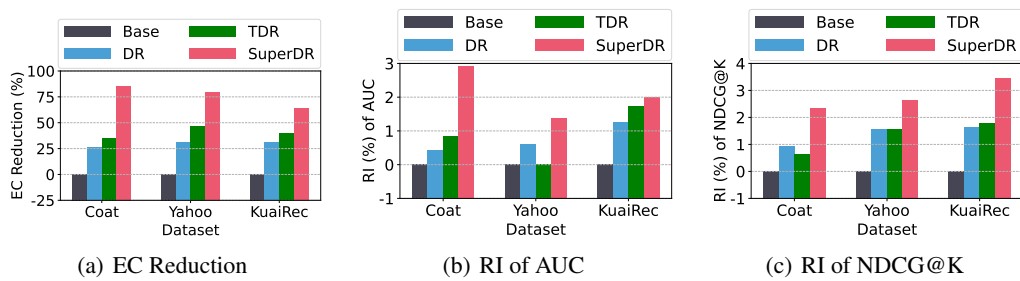

(a) EC Reduction     (b) RI of AUC     (c) RI of NDCG@K

Figure 3: Effects of Empirical Covariance(EC) Reduction (%) on Relative Improvement(RI) (%) of AUC, NDCG@K on three datasets.

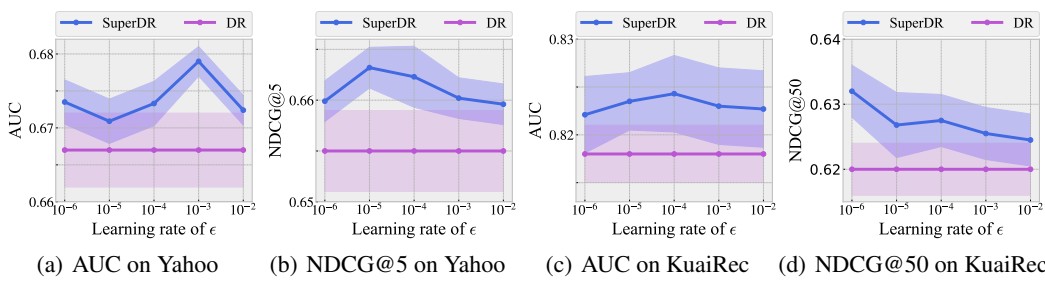

(a) AUC on Yahoo   (b) NDCG@5 on Yahoo   (c) AUC on KuaiRec   (d) NDCG@50 on KuaiRec

Figure 4: Effects of learning rate of correction hyperparameter $\epsilon$ on AUC and NDCG@K.

variance on observed samples. Unfortunately, TDR is unable to control the covariance on missing outcomes, resulting in its performance being inferior to the proposed SuperDR.

## 5.4 SENSITIVITY ANALYSIS

We conduct sensitivity analysis on the **Yahoo** and **KuaiRec** datasets to explore the relationship between the learning rate of learnable parameter $\epsilon$ and the debiasing performance, with AUC and NDCG@K as the evaluation metrics, where K=5 on **Yahoo** and K=50 on **KuaiRec**. As shown in Figure 4, the proposed SuperDR stably outperforms vallina DR under varying learning rates of $\epsilon$, demonstrating that the enhanced imputation model with target learning mitigates the additional bias introduced by sampling and exhibits no-harm property. Meanwhile, under relatively moderate learning rates ($1e-5$, $1e-3$), the SuperDR model demonstrates competitive prediction performance. These results indicate the effectiveness of SuperDR in addressing sampling bias.

## 6 CONCLUSION

In this paper, we addressed the critical issue of selection bias in recommender systems when users and items in the training and test sets are sampled from a larger superpopulation. We demonstrated that traditional doubly robust methods, though effective under certain unbiasedness conditions under a finite population, are biased when the training and test sets do not contain exactly the same users and items even if the imputed errors and learned propensities are correct. To overcome this limitation, we introduced a novel approach, Superpopulation Doubly Robust Target Learning (SuperDR), which is underpinned by a comprehensive theoretical framework. Specifically, we first derive the bias in existing doubly robust estimators has two terms: a covariance term and a term that measures the accuracy of learned propensities and imputed errors. Then we establish new conditions for unbiasedness in the superpopulation scenario. Moreover, we derived a generalization error bound for SuperDR, demonstrating the practical applicability in terms of unbiased learning. In addition, we conducted extensive experiments on three real-world datasets, including a large-scale industrial dataset, and empirically validated the effectiveness of SuperDR in delivering unbiased and accurate recommendations. One of the potential limitations and research direction is how to develop a tighter

bound for control the empirical covariance and to develop a more efficient algorithm for alternatively update the prediction model, the imputation model, and the target learning parameter.

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

## A PROOFS

**Lemma 1** (Bias of DR Estimator (Wang et al., 2019)). *Given imputed errors $\hat{e}_{u,i}$ and learned propensities $\hat{p}_{u,i} > 0$ for all user-item pairs, when considering only the randomness of rating missing indicators, the bias of the DR estimator is*

$$\text{Bias}_{\mathbf{O}}[\mathcal{E}_{\text{DR}}(\theta)] = \frac{1}{|\mathcal{D}|} \sum_{(u,i) \in \mathcal{D}} \frac{\{\hat{p}_{u,i} - p_{u,i}\} \cdot \{e_{u,i} - \hat{e}_{u,i}\}}{\hat{p}_{u,i}}.$$

*Proof of Lemma 1.* The proof can be found in Lemma 3.1 of Wang et al. (2019). However, one should note that, as stated in the proof, "the prediction and imputed errors are treated as constants when taking the expectation, since $o_{u,i}$ does not result from any prediction or imputation models (Schnabel et al., 2016)". The DR estimator in (Wang et al., 2019) is given as

$$\mathcal{E}_{\text{DR}}(\theta) = \frac{1}{|\mathcal{D}|} \sum_{(u,i) \in \mathcal{D}} \left[ \hat{e}_{u,i} + \frac{o_{u,i}(e_{u,i} - \hat{e}_{u,i})}{\hat{p}_{u,i}} \right].$$

By considering only the randomness on $o_{u,i}$, we have

$$\mathbb{E}_{\mathbf{O}}[\mathcal{E}_{\text{DR}}(\theta)] = \mathbb{E}_{\mathbf{O}} \left[ \frac{1}{|\mathcal{D}|} \sum_{(u,i) \in \mathcal{D}} \left[ \hat{e}_{u,i} + \frac{o_{u,i}(e_{u,i} - \hat{e}_{u,i})}{\hat{p}_{u,i}} \right] \right]$$

$$= \frac{1}{|\mathcal{D}|} \sum_{(u,i) \in \mathcal{D}} \left[ \hat{e}_{u,i} + \frac{p_{u,i}(e_{u,i} - \hat{e}_{u,i})}{\hat{p}_{u,i}} \right].$$

By definition, the bias of the DR estimator is

$$\text{Bias}_{\mathbf{O}}[\mathcal{E}_{\text{DR}}(\theta)] = \mathcal{E}_{ideal}(\theta) - \mathbb{E}_{\mathbf{O}}[\mathcal{E}_{\text{DR}}(\theta)]$$

$$= \frac{1}{|\mathcal{D}|} \sum_{(u,i) \in \mathcal{D}} e_{u,i} - \frac{1}{|\mathcal{D}|} \sum_{(u,i) \in \mathcal{D}} \left[ \hat{e}_{u,i} + \frac{p_{u,i}(e_{u,i} - \hat{e}_{u,i})}{\hat{p}_{u,i}} \right]$$

$$= \frac{1}{|\mathcal{D}|} \sum_{(u,i) \in \mathcal{D}} \frac{\{\hat{p}_{u,i} - p_{u,i}\} \cdot \{e_{u,i} - \hat{e}_{u,i}\}}{\hat{p}_{u,i}},$$

which yields the stated results. $\qquad\square$

**Corollary 1** (Double Robustness (Wang et al., 2019)). *The DR estimator is unbiased when either imputed errors $\hat{e}_{u,i}$ or learned propensities $\hat{p}_{u,i}$ are accurate for all user-item pairs, i.e., either $\hat{e}_{u,i} = e_{u,i}$ or $\hat{p}_{u,i} = p_{u,i}$.*

*Proof of Corollary 1.* The proof can be found at Corollary 3.1 in Appendix of (Wang et al., 2019). However, one should note that, as stated in the proof, "the prediction and imputed errors are treated as constants when taking the expectation, since $o_{u,i}$ does not result from any prediction or imputation models (Schnabel et al., 2016)".

Let $\delta_{u,i} = e_{u,i} - \hat{e}_{u,i}$ and $\Delta_{u,i} = \frac{\hat{p}_{u,i} - p_{u,i}}{\hat{p}_{u,i}}$. On the hand, when imputed errors are accurate, we have $\delta_{u,i} = 0$ for $(u,i) \in \mathcal{D}$. In such case, we can compute the bias of the DR estimator by

$$\text{Bias}_{\mathbf{O}}[\mathcal{E}_{\text{DR}}(\theta)] = \frac{1}{|\mathcal{D}|} \sum_{u,i \in \mathcal{D}} \Delta_{u,i} \delta_{u,i} = \frac{1}{|\mathcal{D}|} \sum_{u,i \in \mathcal{D}} \Delta_{u,i} \cdot 0 = 0.$$

On the other hand, when the learned propensities are accurate, we have $\Delta_{u,i} = 0$ for $(u,i) \in \mathcal{D}$. In this case, we can compute the bias of the DR estimator by

$$\text{Bias}(\mathcal{E}_{\text{DR}}) = \frac{1}{|\mathcal{D}|} \sum_{u,i \in \mathcal{D}} \Delta_{u,i} \delta_{u,i} = \frac{1}{|\mathcal{D}|} \sum_{u,i \in \mathcal{D}} 0 \cdot \delta_{u,i} = 0.$$

In both cases, the bias of the DR estimator is zero, which means that the expectation of the DR estimator over all the possible instances of $o_{u,i}$ is exactly the same as the prediction inaccuracy. This completes the proof. $\qquad\square$

**Theorem 1** (Bias of DR Estimator under **Superpopulation**). *Given error imputation model $\hat{e}_{u,i}$ and probabilistic propensity model $\hat{p}_{u,i}$, consider all variables are random, then the bias of the DR estimator, namely* $\mathrm{Bias}[\mathcal{E}_{\mathrm{DR}}(\theta)]$, *is*

$$
\underbrace{\mathrm{Cov}\left(\frac{\hat{p}_{u,i} - o_{u,i}}{\hat{p}_{u,i}}, e_{u,i} - \hat{e}_{u,i}\right)}_{\textit{equals to 0 if independent}} + \underbrace{\mathbb{E}\left[\left\{1 - \mathbb{E}\left[\frac{o_{u,i}}{\hat{p}_{u,i}}\Big| x_{u,i}\right]\right\} \cdot \{\mathbb{E}[e_{u,i} \mid x_{u,i}] - \mathbb{E}[\hat{e}_{u,i} \mid x_{u,i}]\}\right]}_{\textit{equals to 0 either } \mathbb{E}[o_{u,i}/\hat{p}_{u,i} \mid x_{u,i}] = 1 \textit{ or } \mathbb{E}[\hat{e}_{u,i} - e_{u,i} \mid x_{u,i}] = 0}
$$

*Proof of Theorem 1.* Instead of considering only the randomness of the rating missing indicator, in the following, we treat all variables, including imputed errors and learned propensities, as random variables. Formally, we have

$$
\mathrm{Bias}[\mathcal{E}_{\mathrm{DR}}(\theta)] = \mathbb{E}[\mathcal{E}_{ideal}(\theta)] - \mathbb{E}[\mathcal{E}_{\mathrm{DR}}(\theta)] = \mathbb{E}[e_{u,i}] - \mathbb{E}\left[e_{u,i} + \frac{\{o_{u,i} - \hat{p}_{u,i}\} \cdot \{e_{u,i} - \hat{e}_{u,i}\}}{\hat{p}_{u,i}}\right]
$$

$$
= \mathbb{E}\left[\mathbb{E}\left[\left\{\frac{\hat{p}_{u,i} - o_{u,i}}{\hat{p}_{u,i}}\right\}\{e_{u,i} - \hat{e}_{u,i}\} \mid x_{u,i}\right]\right] \qquad \text{(by the double expectation formula)}
$$

$$
= \mathbb{E}\left[\mathbb{E}\left[\left\{\frac{\hat{p}_{u,i} - o_{u,i}}{\hat{p}_{u,i}} - \mathbb{E}\left[\frac{\hat{p}_{u,i} - o_{u,i}}{\hat{p}_{u,i}}\right] + \mathbb{E}\left[\frac{\hat{p}_{u,i} - o_{u,i}}{\hat{p}_{u,i}}\right]\right\}\{(e_{u,i} - \hat{e}_{u,i}) - \mathbb{E}[e_{u,i} - \hat{e}_{u,i}] + \mathbb{E}[e_{u,i} - \hat{e}_{u,i}]\} \mid x_{u,i}\right]\right]
$$

$$
= \mathbb{E}\left[\mathbb{E}\left[\left\{\frac{\hat{p}_{u,i} - o_{u,i}}{\hat{p}_{u,i}} - \mathbb{E}\left[\frac{\hat{p}_{u,i} - o_{u,i}}{\hat{p}_{u,i}}\right]\right\}\{(e_{u,i} - \hat{e}_{u,i}) - \mathbb{E}[e_{u,i} - \hat{e}_{u,i}]\} \mid x_{u,i}\right]\right]
$$

$$
+ \mathbb{E}\left[\left\{1 - \mathbb{E}\left[\frac{o_{u,i}}{\hat{p}_{u,i}}\Big| x_{u,i}\right]\right\} \cdot \{\mathbb{E}[e_{u,i} \mid x_{u,i}] - \mathbb{E}[\hat{e}_{u,i} \mid x_{u,i}]\}\right]
$$

$$
= \mathrm{Cov}\left(\frac{\hat{p}_{u,i} - o_{u,i}}{\hat{p}_{u,i}}, e_{u,i} - \hat{e}_{u,i}\right) + \mathbb{E}\left[\left\{1 - \mathbb{E}\left[\frac{o_{u,i}}{\hat{p}_{u,i}}\Big| x_{u,i}\right]\right\} \cdot \{\mathbb{E}[e_{u,i} \mid x_{u,i}] - \mathbb{E}[\hat{e}_{u,i} \mid x_{u,i}]\}\right],
$$

which yields the stated results. $\qquad\square$

**Corollary 2** (Double Robustness under **Superpopulation**). *The DR estimator is unbiased when both the following conditions hold:*

*(i) conditional independence condition holds, i.e.,* $\mathrm{Cov}((\hat{p}_{u,i} - o_{u,i})/\hat{p}_{u,i}, e_{u,i} - \hat{e}_{u,i}) = 0$;

*(ii) either learned propensities satisfy* $\mathbb{E}[o_{u,i}/\hat{p}_{u,i} \mid x_{u,i}] = 1$, *or imputed errors have the same conditional expectation with true prediction errors* $\mathbb{E}[\hat{e}_{u,i} \mid x_{u,i}] = \mathbb{E}[e_{u,i} \mid x_{u,i}]$.

*Proof of Corollary 2.* First, when condition (i) holds, i.e.,

$$
\mathrm{Cov}((\hat{p}_{u,i} - o_{u,i})/\hat{p}_{u,i}, e_{u,i} - \hat{e}_{u,i}) = 0,
$$

it follows from the results in Theorem 1 that

$$
\mathrm{Bias}[\mathcal{E}_{\mathrm{DR}}(\theta)] = \mathbb{E}\left[\left\{1 - \mathbb{E}\left[\frac{o_{u,i}}{\hat{p}_{u,i}}\Big| x_{u,i}\right]\right\} \cdot \{\mathbb{E}[e_{u,i} \mid x_{u,i}] - \mathbb{E}[\hat{e}_{u,i} \mid x_{u,i}]\}\right]
$$

On the hand, when the learned propensities satisfy $\mathbb{E}[o_{u,i}/\hat{p}_{u,i} \mid x_{u,i}] = 1$. In such case, we can compute the bias of the DR estimator by

$$
\mathrm{Bias}[\mathcal{E}_{\mathrm{DR}}(\theta)] = \mathbb{E}[0 \cdot \{\mathbb{E}[e_{u,i} \mid x_{u,i}] - \mathbb{E}[\hat{e}_{u,i} \mid x_{u,i}]\}] = 0.
$$

On the other hand, when imputed errors have the same conditional expectation with true prediction errors, we have $\mathbb{E}[\hat{e}_{u,i} \mid x_{u,i}] = \mathbb{E}[e_{u,i} \mid x_{u,i}]$. In this case, we can compute the bias of the DR estimator by

$$
\mathrm{Bias}[\mathcal{E}_{\mathrm{DR}}(\theta)] = \mathbb{E}\left[\left\{1 - \mathbb{E}\left[\frac{o_{u,i}}{\hat{p}_{u,i}}\Big| x_{u,i}\right]\right\} \cdot 0\right] = 0.
$$

In both cases, the bias of the DR estimator is zero, which completes the proof. $\qquad\square$

**Definition 1** (Empirical Covariance). *The empirical expected conditional covariance between* $(\hat{p}_{u,i} - o_{u,i})/\hat{p}_{u,i}$ *and* $e_{u,i} - \hat{e}_{u,i}$ *is*

$$\widehat{\text{Cov}}\left(\frac{\hat{p}_{u,i} - o_{u,i}}{\hat{p}_{u,i}}, e_{u,i} - \hat{e}_{u,i}\right) = \frac{1}{|\mathcal{D}|} \sum_{(u,i) \in \mathcal{D}} \frac{\hat{p}_{u,i} - o_{u,i}}{\hat{p}_{u,i}} \cdot (e_{u,i} - \hat{e}_{u,i}).$$

**Definition 2** (Empirical Rademacher Complexity (Shalev-Shwartz and Ben-David, 2014)). *Let* $\mathcal{F}$ *be a family of prediction models mapping from* $x \in \mathcal{X}$ *to* $[a, b]$*, and* $S = \{x_{u,i} \mid (u,i) \in \mathcal{D}\}$ *a fixed sample of size* $|\mathcal{D}|$ *with elements in* $\mathcal{X}$*. Then, the empirical Rademacher complexity of* $\mathcal{F}$ *with respect to the sample* $S$ *is defined as:*

$$\mathcal{R}(\mathcal{F}) = \mathbb{E}_{\boldsymbol{\sigma} \sim \{-1,+1\}^{|\mathcal{D}|}} \sup_{f_\theta \in \mathcal{F}} \left[\frac{1}{|\mathcal{D}|} \sum_{(u,i) \in \mathcal{D}} \sigma_{u,i} e_{u,i}\right],$$

*where* $\boldsymbol{\sigma} = \{\sigma_{u,i} : (u,i) \in \mathcal{D}\}$*, and* $\sigma_{u,i}$ *are independent uniform random variables taking values in* $\{-1, +1\}$*. The random variables* $\sigma_{u,i}$ *are called Rademacher variables.*

**Lemma 2** (Rademacher Comparison Lemma (Shalev-Shwartz and Ben-David, 2014)). *Let* $\mathcal{F}$ *be a family of real-valued functions on* $z \in \mathcal{Z}$ *to* $[a, b]$*, and* $S = \{x_{u,i} \mid (u,i) \in \mathcal{D}\}$ *a fixed sample of size* $|\mathcal{D}|$ *with elements in* $\mathcal{X}$*. Then*

$$\mathbb{E}_{S \sim \mathbb{P}^{|\mathcal{D}|}} \left[\sup_{f \in \mathcal{F}} \left(\mathbb{E}_{z \sim \mathbb{P}}[f(z)] - \frac{1}{|\mathcal{D}|} \sum_{(u,i) \in \mathcal{D}} f(z_{u,i})\right)\right] \leq 2 \mathbb{E}_{S \sim \mathbb{P}^{|\mathcal{D}|}} \mathbb{E}_{\boldsymbol{\sigma} \sim \{-1,+1\}^{|\mathcal{D}|}} \sup_{f \in \mathcal{F}} \left[\frac{1}{|\mathcal{D}|} \sum_{(u,i) \in \mathcal{D}} \sigma_{u,i} f(z_{u,i})\right],$$

*where* $\boldsymbol{\sigma} = \{\sigma_{u,i} : (u,i) \in \mathcal{D}\}$*, and* $\sigma_{u,i}$ *are independent uniform random variables taking values in* $\{-1, +1\}$*. The random variables* $\sigma_{u,i}$ *are called Rademacher variables.*

*Proof of Lemma 2.* The proof can be found in Lemma 26.2 of (Shalev-Shwartz and Ben-David, 2014). $\square$

**Lemma 3** (McDiarmid's Inequality (Shalev-Shwartz and Ben-David, 2014)). *Let* $V$ *be some set and let* $f : V^m \to \mathbb{R}$ *be a function of* $m$ *variables such that for some* $c > 0$*, for all* $i \in [m]$ *and for all* $x_1, \ldots, x_m, x_i' \in V$ *we have*

$$|f(x_1, \ldots, x_m) - f(x_1, \ldots, x_{i-1}, x_i', x_{i+1}, \ldots, x_m)| \leq c$$

*Let* $X_1, \ldots, X_m$ *be* $m$ *independent random variables taking values in* $V$*. Then, with probability of at least* $1 - \delta$ *we have*

$$|f(X_1, \ldots, X_m) - \mathbb{E}[f(X_1, \ldots, X_m)]| \leq c\sqrt{\log\left(\frac{2}{\delta}\right) m/2}$$

*Proof of Lemma 3.* The proof can be found in Lemma 26.4 of (Shalev-Shwartz and Ben-David, 2014). $\square$

**Lemma 4** (Rademacher Calculus (Shalev-Shwartz and Ben-David, 2014)). *For any* $A \subset \mathbb{R}^m$*, scalar* $c \in \mathbb{R}$*, and vector* $\mathbf{a}_0 \in \mathbb{R}^m$*, we have*

$$R(\{c\mathbf{a} + \mathbf{a}_0 : \mathbf{a} \in A\}) \leq |c|R(A).$$

*Proof of Lemma 4.* The proof can be found in Lemma 26.6 of (Shalev-Shwartz and Ben-David, 2014). $\square$

**Theorem 2** (Controllability of Empirical Covariance). *The boosted imputation model trained by minimizing the balanced enhanced imputation loss is sufficient for controlling the empirical covariance.*

*(i) For user-item pairs with **observed** outcomes, the empirical covariance is 0. Formally, we have*

$$\frac{\partial}{\partial \epsilon} \mathcal{L}_e^{Bal}(\phi, \epsilon) \bigg|_{\epsilon = \epsilon^*} = 0, \quad \text{which is equivalent to} \quad \frac{1}{|\mathcal{D}|} \sum_{(u,i):\, o_{u,i}=1} \frac{\hat{p}_{u,i} - o_{u,i}}{\hat{p}_{u,i}} \cdot (e_{u,i} - \tilde{e}_{u,i}) = 0;$$

*(ii) For user-item pairs with **missing** outcomes, suppose that $\hat{p}_{u,i} \geq K_\psi$ and $|e_{u,i} - \hat{e}_{u,i}| \leq K_\phi$, then with probability at least $1 - \eta$, we have*

$$\frac{1}{|\mathcal{D}|} \sum_{(u,i):\, o_{u,i}=0} \frac{\hat{p}_{u,i} - o_{u,i}}{\hat{p}_{u,i}} \cdot (e_{u,i} - \tilde{e}_{u,i}) \leq \underbrace{\mathcal{L}_e^{Bal}(\phi, \epsilon)^{\frac{1}{2}}}_{\text{proposed loss}} + K_\phi \cdot \underbrace{\left[ \frac{1}{|\mathcal{D}|} \sum_{u,i \in \mathcal{D}} \left| 1 - \mathbb{E}\left[ \frac{o_{u,i}}{\hat{p}_{u,i}} \Big| x_{u,i} \right] \right| \right]^{\frac{1}{2}}}_{\text{empirical bias from probabilistic propensity model}}$$

$$+ \underbrace{\left[ K_\phi \left( 1 + \frac{1}{K_\psi} \right) \left( 2\mathcal{R}(\mathcal{F}) + K_\phi \sqrt{\frac{18 \log(4/\eta)}{|\mathcal{D}|}} \right) \right]^{\frac{1}{2}}}_{\text{tail bound controlled by empirical Rademacher complexity and sample size}}.$$

*Proof.* For the proof of Theorem 2(i), first recap that the proposed boosted imputation model is

$$\tilde{e}_{u,i} = m(x_{u,i}; \phi) + \epsilon(o_{u,i} - \pi(x_{u,i}; \psi)),$$

and the proposed balancing enhanced loss function for training the boosted imputation model is

$$(\phi^*, \epsilon^*) = \arg\min_{\phi, \epsilon} \mathcal{L}_e^{Bal}(\phi, \epsilon) = \frac{1}{|\mathcal{D}|} \sum_{(u,i)\in\mathcal{D}} \frac{o_{u,i}(e_{u,i} - \tilde{e}_{u,i})^2}{\hat{p}_{u,i}}.$$

By taking the partial derivative with respective to $\epsilon$ of the above formula and setting it to zero, we have

$$\frac{\partial}{\partial \epsilon} \mathcal{L}_e^{Bal}(\phi, \epsilon) \bigg|_{\epsilon = \epsilon^*} = 0, \quad \text{which is equivalent to} \quad \frac{1}{|\mathcal{D}|} \sum_{(u,i):\, o_{u,i}=1} \frac{\hat{p}_{u,i} - o_{u,i}}{\hat{p}_{u,i}} \cdot (e_{u,i} - \tilde{e}_{u,i}) = 0,$$

which proves the empirical convariance on the observed outcomes is 0.

For the proof of Theorem 2(ii), by noting that

$$\frac{1}{|\mathcal{D}|} \sum_{(u,i):\, o_{u,i}=0} \frac{\hat{p}_{u,i} - o_{u,i}}{\hat{p}_{u,i}} \cdot (e_{u,i} - \tilde{e}_{u,i}) = \frac{1}{|\mathcal{D}|} \sum_{(u,i):\, o_{u,i}=0} (e_{u,i} - \tilde{e}_{u,i}) \leq \left[ \frac{1}{|\mathcal{D}|} \sum_{(u,i)\in\mathcal{D}} (e_{u,i} - \tilde{e}_{u,i})^2 \right]^{\frac{1}{2}},$$

we now focus on bounding the last term of the above equation with the least probability.

Suppose that $\hat{p}_{u,i} \geq K_\psi$ and $|e_{u,i} - \tilde{e}_{u,i}| \leq K_\phi$, then

$$\frac{1}{|\mathcal{D}|} \sum_{(u,i)\in\mathcal{D}} (e_{u,i} - \tilde{e}_{u,i})^2 = \frac{1}{|\mathcal{D}|} \sum_{(u,i)\in\mathcal{D}} \frac{o_{u,i}(e_{u,i} - \tilde{e}_{u,i})^2}{\hat{p}_{u,i}} + \frac{1}{|\mathcal{D}|} \sum_{(u,i)\in\mathcal{D}} (e_{u,i} - \tilde{e}_{u,i})^2$$

$$- \mathbb{E}\left[ \frac{1}{|\mathcal{D}|} \sum_{(u,i)\in\mathcal{D}} \frac{o_{u,i}(e_{u,i} - \tilde{e}_{u,i})^2}{\hat{p}_{u,i}} \right] + \mathbb{E}\left[ \frac{1}{|\mathcal{D}|} \sum_{(u,i)\in\mathcal{D}} \frac{o_{u,i}(e_{u,i} - \tilde{e}_{u,i})^2}{\hat{p}_{u,i}} \right] - \frac{1}{|\mathcal{D}|} \sum_{(u,i)\in\mathcal{D}} \frac{o_{u,i}(e_{u,i} - \tilde{e}_{u,i})^2}{\hat{p}_{u,i}}$$

$$\leq \frac{1}{|\mathcal{D}|} \sum_{(u,i)\in\mathcal{D}} \frac{o_{u,i}(e_{u,i} - \tilde{e}_{u,i})^2}{\hat{p}_{u,i}} + \left| \frac{1}{|\mathcal{D}|} \sum_{(u,i)\in\mathcal{D}} (e_{u,i} - \tilde{e}_{u,i})^2 - \mathbb{E}\left[ \frac{1}{|\mathcal{D}|} \sum_{(u,i)\in\mathcal{D}} \frac{o_{u,i}(e_{u,i} - \tilde{e}_{u,i})^2}{\hat{p}_{u,i}} \right] \right|$$

$$+ \left( \mathbb{E}\left[ \frac{1}{|\mathcal{D}|} \sum_{(u,i)\in\mathcal{D}} \frac{o_{u,i}(e_{u,i} - \tilde{e}_{u,i})^2}{\hat{p}_{u,i}} \right] - \frac{1}{|\mathcal{D}|} \sum_{(u,i)\in\mathcal{D}} \frac{o_{u,i}(e_{u,i} - \tilde{e}_{u,i})^2}{\hat{p}_{u,i}} \right)$$

$$\leq \mathcal{L}_e^{Bal}(\phi,\epsilon) + K_\phi^2 \cdot \left| \mathbb{E}\left[ \frac{1}{|\mathcal{D}|} \sum_{(u,i)\in\mathcal{D}} 1 - \frac{o_{u,i}}{\hat{p}_{u,i}} \right] \right|$$

$$+ \sup_{f_\theta \in \mathcal{F}} \left( \mathbb{E}\left[ \frac{1}{|\mathcal{D}|} \sum_{(u,i)\in\mathcal{D}} \frac{o_{u,i}(e_{u,i} - \tilde{e}_{u,i})^2}{\hat{p}_{u,i}} \right] - \frac{1}{|\mathcal{D}|} \sum_{(u,i)\in\mathcal{D}} \frac{o_{u,i}(e_{u,i} - \tilde{e}_{u,i})^2}{\hat{p}_{u,i}} \right).$$

For simplicity, we denote the last term in the above formula as

$$\mathcal{B}(\mathcal{F}) = \sup_{f_\theta \in \mathcal{F}} \left( \mathbb{E}\left[ \frac{1}{|\mathcal{D}|} \sum_{(u,i)\in\mathcal{D}} \frac{o_{u,i}(e_{u,i} - \tilde{e}_{u,i})^2}{\hat{p}_{u,i}} \right] - \frac{1}{|\mathcal{D}|} \sum_{(u,i)\in\mathcal{D}} \frac{o_{u,i}(e_{u,i} - \tilde{e}_{u,i})^2}{\hat{p}_{u,i}} \right),$$

we then aim to bound $\mathcal{B}(\mathcal{F})$ in the following.

Note that

$$\mathcal{B}(\mathcal{F}) = \mathop{\mathbb{E}}_{S \sim \mathbb{P}^{|\mathcal{D}|}}[\mathcal{B}(\mathcal{F})] + \left\{ \mathcal{B}(\mathcal{F}) - \mathop{\mathbb{E}}_{S \sim \mathbb{P}^{|\mathcal{D}|}}[\mathcal{B}(\mathcal{F})] \right\},$$

where the first term is $\mathop{\mathbb{E}}_{S \sim \mathbb{P}^{|\mathcal{D}|}}[\mathcal{B}(\mathcal{F})]$, and by Lemma 2 we have

$$\mathop{\mathbb{E}}_{S \sim \mathbb{P}^{|\mathcal{D}|}}[\mathcal{B}(\mathcal{F})] \leq 2 \mathop{\mathbb{E}}_{S \sim \mathbb{P}^{|\mathcal{D}|}} \mathbb{E}_{\boldsymbol{\sigma} \sim \{-1,+1\}^{|\mathcal{D}|}} \sup_{f_\theta \in \mathcal{F}} \left[ \frac{1}{|\mathcal{D}|} \sum_{(u,i)\in\mathcal{D}} \sigma_{u,i} \frac{o_{u,i}(e_{u,i} - \tilde{e}_{u,i})^2}{\hat{p}_{u,i}} \right].$$

By the assumptions that $\hat{p}_{u,i} \geq K_\psi$ and $|e_{u,i} - \tilde{e}_{u,i}| \leq K_\phi$, we have

$$\mathop{\mathbb{E}}_{S \sim \mathbb{P}^{|\mathcal{D}|}}[\mathcal{B}(\mathcal{F})] \leq 2K_\phi \left( 1 + \frac{1}{K_\psi} \right) \mathop{\mathbb{E}}_{S \sim \mathbb{P}^{|\mathcal{D}|}} \mathbb{E}_{\boldsymbol{\sigma} \sim \{-1,+1\}^{|\mathcal{D}|}} \sup_{f_\theta \in \mathcal{F}} \left[ \frac{1}{|\mathcal{D}|} \sum_{(u,i)\in\mathcal{D}} \sigma_{u,i}(e_{u,i} - \tilde{e}_{u,i}) \right]$$

$$= 2K_\phi \left( 1 + \frac{1}{K_\psi} \right) \mathop{\mathbb{E}}_{S \sim \mathbb{P}^{|\mathcal{D}|}}\{\mathcal{R}(\mathcal{F})\},$$

where the last equation is directly from Lemma 4, and $\mathcal{R}(\mathcal{F})$ is the empirical Rademacher complexity

$$\mathcal{R}(\mathcal{F}) = \mathbb{E}_{\boldsymbol{\sigma} \sim \{-1,+1\}^{|\mathcal{D}|}} \sup_{f_\theta \in \mathcal{F}} \left[ \frac{1}{|\mathcal{D}|} \sum_{(u,i)\in\mathcal{D}} \sigma_{u,i} e_{u,i} \right],$$

where $\boldsymbol{\sigma} = \{\sigma_{u,i} : (u,i) \in \mathcal{D}\}$, and $\sigma_{u,i}$ are independent uniform random variables taking values in $\{-1,+1\}$. The random variables $\sigma_{u,i}$ are called Rademacher variables.

By applying McDiarmid's inequality in Lemma 3, and let $c = \frac{2K_\phi}{|\mathcal{D}|}$, with probability at least $1 - \frac{\eta}{2}$,

$$\left| \mathcal{R}(\mathcal{F}) - \mathop{\mathbb{E}}_{S \sim \mathbb{P}^{|\mathcal{D}|}}\{\mathcal{R}(\mathcal{F})\} \right| \leq 2K_\phi \sqrt{\frac{\log(4/\eta)}{2|\mathcal{D}|}} = K_\phi \sqrt{\frac{2\log(4/\eta)}{|\mathcal{D}|}}.$$

For the rest term $\mathcal{B}(\mathcal{F}) - \mathop{\mathbb{E}}\limits_{S \sim \mathbb{P}^{|\mathcal{D}|}}[\mathcal{B}(\mathcal{F})]$, by applying McDiarmid's inequality in Lemma 3 and the

assumptions that $\hat{p}_{u,i} \geq K_\psi$ and $|e_{u,i} - \tilde{e}_{u,i}| \leq K_\phi$, let $c = \frac{2K_\phi^2\left(1+\frac{1}{K_\psi}\right)}{|\mathcal{D}|}$, then with probability at least $1 - \frac{\eta}{2}$,

$$\left| \mathcal{B}(\mathcal{F}) - \mathop{\mathbb{E}}\limits_{S \sim \mathbb{P}^{|\mathcal{D}|}}[\mathcal{B}(\mathcal{F})] \right| \leq 2K_\phi^2 \left(1 + \frac{1}{K_\psi}\right) \sqrt{\frac{\log(4/\eta)}{2|\mathcal{D}|}} = K_\phi^2 \left(1 + \frac{1}{K_\psi}\right) \sqrt{\frac{2\log(4/\eta)}{|\mathcal{D}|}}.$$

We now bound $\mathcal{B}(\mathcal{F})$ combining the above results. Formally, we have

$$\mathcal{B}(\mathcal{F}) = \mathop{\mathbb{E}}\limits_{S \sim \mathbb{P}^{|\mathcal{D}|}}[\mathcal{B}(\mathcal{F})] + \left\{ \mathcal{B}(\mathcal{F}) - \mathop{\mathbb{E}}\limits_{S \sim \mathbb{P}^{|\mathcal{D}|}}[\mathcal{B}(\mathcal{F})] \right\}$$

$$\leq 2K_\phi \left(1 + \frac{1}{K_\psi}\right) \mathop{\mathbb{E}}\limits_{S \sim \mathbb{P}^{|\mathcal{D}|}}\{\mathcal{R}(\mathcal{F})\} + \left\{ \mathcal{B}(\mathcal{F}) - \mathop{\mathbb{E}}\limits_{S \sim \mathbb{P}^{|\mathcal{D}|}}[\mathcal{B}(\mathcal{F})] \right\}.$$

With probability at least $1 - \eta$, we have

$$\mathcal{B}(\mathcal{F}) \leq 2K_\phi \left(1 + \frac{1}{K_\psi}\right) \left( \mathcal{R}(\mathcal{F}) + K_\phi \sqrt{\frac{2\log(4/\eta)}{|\mathcal{D}|}} \right) + K_\phi^2 \left(1 + \frac{1}{K_\psi}\right) \sqrt{\frac{2\log(4/\eta)}{|\mathcal{D}|}}$$

$$= K_\phi \left(1 + \frac{1}{K_\psi}\right) \left( 2\mathcal{R}(\mathcal{F}) + K_\phi \sqrt{\frac{18\log(4/\eta)}{|\mathcal{D}|}} \right).$$

We now bound the empirical convariance on the missing outcomes combining the above results. Formally, we have

$$\frac{1}{|\mathcal{D}|} \sum_{(u,i):\, o_{u,i}=0} \frac{\hat{p}_{u,i} - o_{u,i}}{\hat{p}_{u,i}} \cdot (e_{u,i} - \tilde{e}_{u,i}) \leq \left[ \frac{1}{|\mathcal{D}|} \sum_{(u,i)\in\mathcal{D}} (e_{u,i} - \tilde{e}_{u,i})^2 \right]^{\frac{1}{2}}$$

$$\leq \left[ \mathcal{L}_e^{Bal}(\phi,\epsilon) + \frac{K_\phi^2}{|\mathcal{D}|} \sum_{u,i\in\mathcal{D}} \left| 1 - \mathbb{E}\left[ \frac{o_{u,i}}{\hat{p}_{u,i}} \Big| x_{u,i} \right] \right| + K_\phi \left(1 + \frac{1}{K_\psi}\right) \left( 2\mathcal{R}(\mathcal{F}) + K_\phi \sqrt{\frac{18\log(4/\eta)}{|\mathcal{D}|}} \right) \right]^{\frac{1}{2}}$$

$$\leq \mathcal{L}_e^{Bal}(\phi,\epsilon)^{\frac{1}{2}} + K_\phi \cdot \left[ \frac{1}{|\mathcal{D}|} \sum_{u,i\in\mathcal{D}} \left| 1 - \mathbb{E}\left[ \frac{o_{u,i}}{\hat{p}_{u,i}} \Big| x_{u,i} \right] \right| \right]^{\frac{1}{2}} +$$

$$\left[ K_\phi \left(1 + \frac{1}{K_\psi}\right) \left( 2\mathcal{R}(\mathcal{F}) + K_\phi \sqrt{\frac{18\log(4/\eta)}{|\mathcal{D}|}} \right) \right]^{\frac{1}{2}},$$

which yields the stated results. $\qquad\square$

**Corollary 3** (Relation to previous imputed errors). *The learned coefficient $\epsilon^*$ will converge to zero when the probabilistic imputation model $\hat{e}_{u,i}$ has already led to zero empirical covariance, making $\tilde{e}_{u,i}$ degenerates to $\hat{e}_{u,i}$.*

*Proof of Corollary 3.* Note that $\epsilon^*$ is solved by minimizing

$$\frac{1}{|\mathcal{D}|} \sum_{(u,i)\in\mathcal{D}} \frac{o_{u,i}(e_{u,i} - \hat{e}_{u,i} - \epsilon(o_{u,i} - \hat{p}_{u,i}))^2}{\hat{p}_{u,i}}.$$

Taking the first derivative of the above loss with respect to $\epsilon$ and setting it to zero yields

$$\sum_{(u,i)\in\mathcal{D}} \frac{o_{u,i}}{\hat{p}_{u,i}} \cdot \left\{ e_{u,i} - \hat{e}_{u,i} - \epsilon(o_{u,i} - \hat{p}_{u,i}) \right\} \cdot (o_{u,i} - \hat{p}_{u,i}) = 0,$$

which implies that

$$\sum_{(u,i)\in\mathcal{D}} \frac{o_{u,i}}{\hat{p}_{u,i}} \cdot \{e_{u,i} - \tilde{e}_{u,i}\} \cdot (o_{u,i} - \hat{p}_{u,i}) = 0,$$

from which implies the uniqueness of $\epsilon$. Formally, if $\hat{e}_{u,i}$ already satisfies zero empirical covariance on the observed outcomes, then $\epsilon = 0$ is a solution of the above equation. Let $\hat{\epsilon}$ be another solution of the above equation. Since the solution of equation is unique, then $\hat{\epsilon}$ will converage to 0, making $\tilde{e}_{u,i}$ degenerates to $\hat{e}_{u,i}$. $\qquad\square$

**Corollary 4** (Bias reduction property). *The proposed balancing enhanced imputation loss leads to the smaller bias of imputed errors $\tilde{e}_{u,i}$, when $\hat{e}_{u,i}$ are inaccurate. Formally, we have*

$$\min_{\phi,\epsilon} \mathcal{L}_e^{Bal}(\phi,\epsilon) = \frac{1}{|\mathcal{D}|} \sum_{(u,i)\in\mathcal{D}} \frac{o_{u,i}(e_{u,i} - \tilde{e}_{u,i})^2}{\hat{p}_{u,i}} \leq \min_{\phi} \mathcal{L}_e(\phi) = \frac{1}{|\mathcal{D}|} \sum_{(u,i)\in\mathcal{D}} \frac{o_{u,i}(e_{u,i} - \hat{e}_{u,i})^2}{\hat{p}_{u,i}}.$$

*Proof of Corollary 4.* The result holds by noting that

$$\min_{\phi,\epsilon} \mathcal{L}_e^{Bal}(\phi,\epsilon) \leq \min_{\phi} \mathcal{L}_e^{Bal}(\phi,\epsilon=0) = \min_{\phi} \mathcal{L}_e(\phi) = \frac{1}{|\mathcal{D}|} \sum_{(u,i)\in\mathcal{D}} \frac{o_{u,i}(e_{u,i} - \hat{e}_{u,i})^2}{\hat{p}_{u,i}}.$$

$\qquad\square$

**Corollary 5** (Variance reduction property). *The proposed balancing enhanced imputation loss leads to the smaller variance of $\tilde{e}_{u,i}$ when the optimal $\epsilon^*$ lies in a certain range. Formally, we have*

$$\mathbb{V}(\tilde{e}_{u,i}) = \mathbb{V}(\hat{e}_{u,i} + \epsilon^* \cdot (o_{u,i} - \hat{p}_{u,i})) \leq \mathbb{V}(\hat{e}_{u,i}), \quad\text{if}\quad \epsilon^* \in \left[0,\, 2 \cdot \frac{\text{Cov}(\hat{e}_{u,i}, \hat{p}_{u,i} - o_{u,i})}{\mathbb{V}(\hat{p}_{u,i} - o_{u,i})}\right].$$

*Proof of Corollary 5.* First, we note that $\mathbb{V}(\tilde{e}_{u,i})$ equals to

$$\mathbb{V}(\hat{e}_{u,i}) - 2\epsilon^* \text{Cov}(\hat{e}_{u,i}, \hat{p}_{u,i} - o_{u,i}) + (\epsilon^*)^2 \mathbb{V}(o_{u,i} - \hat{p}_{u,i}),$$

which serves as a quadratic function with respect to $\epsilon^*$. By taking the partial derivative respective to $\epsilon^*$ of the above formula and setting it to zero, the optimal $\epsilon^*$ with the minimal variance is given as

$$\epsilon^* = \frac{\text{Cov}(\hat{e}_{u,i}, \hat{p}_{u,i} - o_{u,i})}{\mathbb{V}(\hat{p}_{u,i} - o_{u,i})}.$$

By exploiting the symmetry of the quadratic function, we have

$$\mathbb{V}(\tilde{e}_{u,i}) = \mathbb{V}(\hat{e}_{u,i} + \epsilon^* \cdot (o_{u,i} - \hat{p}_{u,i})) \leq \mathbb{V}(\hat{e}_{u,i}),$$
$$\text{if}\quad \epsilon^* \in \left[0,\, 2 \cdot \frac{\text{Cov}(\hat{e}_{u,i}, \hat{p}_{u,i} - o_{u,i})}{\mathbb{V}(\hat{p}_{u,i} - o_{u,i})}\right].$$

$\qquad\square$

**Theorem 3** (Generalization Bound under Probabilistic Models). *Suppose that $\hat{p}_{u,i} \geq K_\psi$ and $\min\{\hat{e}_{u,i}, |e_{u,i} - \hat{e}_{u,i}|\} \leq K_\phi$, then with probability at least $1 - \eta$, we have*

$$\mathcal{L}_{ideal}(\theta) \leq \underbrace{\mathcal{L}_{\text{DR}}(\theta) + \frac{1}{|\mathcal{D}|} \sum_{(u,i)\in\mathcal{D}} \left|1 - \mathbb{E}\left[\frac{o_{u,i}}{\hat{p}_{u,i}}\Big|x_{u,i}\right]\right| \cdot \left|\mathbb{E}[e_{u,i} \mid x_{u,i}] - \mathbb{E}[\hat{e}_{u,i} \mid x_{u,i}]\right|}_{\text{vanilla DR only controls the empirical DR loss, and empirical risks of imputation and propensity models}}$$

$$+ \underbrace{\left|\frac{1}{|\mathcal{D}|} \sum_{(u,i)\in\mathcal{D}} \text{Cov}\left(\frac{o_{u,i} - \hat{p}_{u,i}}{\hat{p}_{u,i}}, e_{u,i} - \hat{e}_{u,i}\right)\right|}_{\text{balancing enhanced DR further controls the independence}} + \underbrace{\left(1 + \frac{1}{K_\psi}\right)\left(2\mathcal{R}(\mathcal{F}) + K_\phi\sqrt{\frac{18\log(4/\eta)}{|\mathcal{D}|}}\right)}_{\text{tail bound controlled by empirical Rademacher complexity and sample size}}$$

*Proof of Theorem 3.* First we decompose the ideal loss as follows.

$$
\begin{aligned}
\mathcal{L}_{ideal}(\theta) &= \mathcal{L}_{\mathrm{DR}}(\theta) + (\mathcal{L}_{ideal}(\theta) - \mathbb{E}[\mathcal{L}_{\mathrm{DR}}(\theta)]) + (\mathbb{E}[\mathcal{L}_{\mathrm{DR}}(\theta)] - \mathcal{L}_{\mathrm{DR}}(\theta)) \\
&= \mathcal{L}_{\mathrm{DR}}(\theta) + \mathrm{Bias}[\mathcal{L}_{\mathrm{DR}}(\theta)] + (\mathbb{E}[\mathcal{L}_{\mathrm{DR}}(\theta)] - \mathcal{L}_{\mathrm{DR}}(\theta)) \\
&\le \mathcal{L}_{\mathrm{DR}}(\theta) + |\mathrm{Bias}[\mathcal{L}_{\mathrm{DR}}(\theta)]| \\
&\quad + \sup_{f_\theta \in \mathcal{F}} \left( \mathbb{E}\left[ \frac{1}{|\mathcal{D}|} \sum_{(u,i)\in\mathcal{D}} \hat{e}_{u,i} + \frac{o_{u,i}(e_{u,i} - \hat{e}_{u,i})}{\hat{p}_{u,i}} \right] - \frac{1}{|\mathcal{D}|} \sum_{(u,i)\in\mathcal{D}} \hat{e}_{u,i} - \frac{o_{u,i}(e_{u,i} - \hat{e}_{u,i})}{\hat{p}_{u,i}} \right).
\end{aligned}
$$

For simplicity, we denote the last term in the above formula as

$$
\mathcal{B}(\mathcal{F}) = \sup_{f_\theta \in \mathcal{F}} \left( \mathbb{E}\left[ \frac{1}{|\mathcal{D}|} \sum_{(u,i)\in\mathcal{D}} \hat{e}_{u,i} + \frac{o_{u,i}(e_{u,i} - \hat{e}_{u,i})}{\hat{p}_{u,i}} \right] - \frac{1}{|\mathcal{D}|} \sum_{(u,i)\in\mathcal{D}} \hat{e}_{u,i} - \frac{o_{u,i}(e_{u,i} - \hat{e}_{u,i})}{\hat{p}_{u,i}} \right),
$$

we then aim to bound $\mathcal{B}(\mathcal{F})$ in the following.

Note that

$$
\mathcal{B}(\mathcal{F}) = \mathbb{E}_{S \sim \mathbb{P}^{|\mathcal{D}|}}[\mathcal{B}(\mathcal{F})] + \left\{ \mathcal{B}(\mathcal{F}) - \mathbb{E}_{S \sim \mathbb{P}^{|\mathcal{D}|}}[\mathcal{B}(\mathcal{F})] \right\},
$$

where the first term is $\mathbb{E}_{S \sim \mathbb{P}^{|\mathcal{D}|}}[\mathcal{B}(\mathcal{F})]$, and by Lemma 2 we have

$$
\mathbb{E}_{S \sim \mathbb{P}^{|\mathcal{D}|}}[\mathcal{B}(\mathcal{F})] \le 2 \, \mathbb{E}_{S \sim \mathbb{P}^{|\mathcal{D}|}} \mathbb{E}_{\boldsymbol{\sigma} \sim \{-1,+1\}^{|\mathcal{D}|}} \sup_{f_\theta \in \mathcal{F}} \left[ \frac{1}{|\mathcal{D}|} \sum_{(u,i)\in\mathcal{D}} \sigma_{u,i} \hat{e}_{u,i} + \frac{\sigma_{u,i} o_{u,i}(e_{u,i} - \hat{e}_{u,i})}{\hat{p}_{u,i}} \right].
$$

By the assumptions that $\hat{p}_{u,i} \ge K_\psi$ and $\min\{\hat{e}_{u,i}, |e_{u,i} - \hat{e}_{u,i}|\} \le K_\phi$, we have

$$
\mathbb{E}_{S \sim \mathbb{P}^{|\mathcal{D}|}}[\mathcal{B}(\mathcal{F})] \le 2 \, \mathbb{E}_{S \sim \mathbb{P}^{|\mathcal{D}|}} \mathbb{E}_{\boldsymbol{\sigma} \sim \{-1,+1\}^{|\mathcal{D}|}} \sup_{f_\theta \in \mathcal{F}} \left[ \frac{1}{|\mathcal{D}|} \sum_{(u,i)\in\mathcal{D}} \frac{\sigma_{u,i} o_{u,i}(e_{u,i} - \hat{e}_{u,i})}{\hat{p}_{u,i}} \right]
$$

$$
\le 2 \left( 1 + \frac{1}{K_\psi} \right) \mathbb{E}_{S \sim \mathbb{P}^{|\mathcal{D}|}} \{\mathcal{R}(\mathcal{F})\},
$$

where the first equation is from Lemma 4, and $\mathcal{R}(\mathcal{F})$ is the empirical Rademacher complexity

$$
\mathcal{R}(\mathcal{F}) = \mathbb{E}_{\boldsymbol{\sigma} \sim \{-1,+1\}^{|\mathcal{D}|}} \sup_{f_\theta \in \mathcal{F}} \left[ \frac{1}{|\mathcal{D}|} \sum_{(u,i)\in\mathcal{D}} \sigma_{u,i} e_{u,i} \right],
$$

where $\boldsymbol{\sigma} = \{\sigma_{u,i} : (u,i) \in \mathcal{D}\}$, and $\sigma_{u,i}$ are independent uniform random variables taking values in $\{-1,+1\}$. The random variables $\sigma_{u,i}$ are called Rademacher variables.

By applying McDiarmid's inequality in Lemma 3, and let $c = \frac{2K_\phi}{|\mathcal{D}|}$, with probability at least $1 - \frac{\eta}{2}$,

$$
\left| \mathcal{R}(\mathcal{F}) - \mathbb{E}_{S \sim \mathbb{P}^{|\mathcal{D}|}} \{\mathcal{R}(\mathcal{F})\} \right| \le 2K_\phi \sqrt{\frac{\log(4/\eta)}{2|\mathcal{D}|}} = K_\phi \sqrt{\frac{2\log(4/\eta)}{|\mathcal{D}|}}.
$$

For the rest term $\mathcal{B}(\mathcal{F}) - \mathbb{E}_{S \sim \mathbb{P}^{|\mathcal{D}|}}[\mathcal{B}(\mathcal{F})]$, by applying McDiarmid's inequality in Lemma 3 and the assumptions that $\hat{p}_{u,i} \ge K_\psi$ and $\min\{\hat{e}_{u,i}, |e_{u,i} - \hat{e}_{u,i}|\} \le K_\phi$, let $c = \frac{2K_\phi\left(1+\frac{1}{K_\psi}\right)}{|\mathcal{D}|}$, then with probability at least $1 - \frac{\eta}{2}$,

$$
\left| \mathcal{B}(\mathcal{F}) - \mathbb{E}_{S \sim \mathbb{P}^{|\mathcal{D}|}}[\mathcal{B}(\mathcal{F})] \right| \le 2K_\phi \left( 1 + \frac{1}{K_\psi} \right) \sqrt{\frac{\log(4/\eta)}{2|\mathcal{D}|}} = K_\phi \left( 1 + \frac{1}{K_\psi} \right) \sqrt{\frac{2\log(4/\eta)}{|\mathcal{D}|}}.
$$

We now bound $\mathcal{B}(\mathcal{F})$ combining the above results. Formally, we have

$$\mathcal{B}(\mathcal{F}) = \mathop{\mathbb{E}}_{S \sim \mathbb{P}^{|\mathcal{D}|}}[\mathcal{B}(\mathcal{F})] + \left\{ \mathcal{B}(\mathcal{F}) - \mathop{\mathbb{E}}_{S \sim \mathbb{P}^{|\mathcal{D}|}}[\mathcal{B}(\mathcal{F})] \right\}$$

$$\leq 2 \left( 1 + \frac{1}{K_\psi} \right) \mathop{\mathbb{E}}_{S \sim \mathbb{P}^{|\mathcal{D}|}} \{\mathcal{R}(\mathcal{F})\} + \left\{ \mathcal{B}(\mathcal{F}) - \mathop{\mathbb{E}}_{S \sim \mathbb{P}^{|\mathcal{D}|}}[\mathcal{B}(\mathcal{F})] \right\}.$$

With probability at least $1 - \eta$, we have

$$\mathcal{B}(\mathcal{F}) \leq 2 \left( 1 + \frac{1}{K_\psi} \right) \left( \mathcal{R}(\mathcal{F}) + K_\phi \sqrt{\frac{2\log(4/\eta)}{|\mathcal{D}|}} \right) + K_\phi \left( 1 + \frac{1}{K_\psi} \right) \sqrt{\frac{2\log(4/\eta)}{|\mathcal{D}|}}$$

$$= \left( 1 + \frac{1}{K_\psi} \right) \left( 2\mathcal{R}(\mathcal{F}) + K_\phi \sqrt{\frac{18\log(4/\eta)}{|\mathcal{D}|}} \right).$$

We now bound the ideal loss combining the above results. Formally, we have

$$\mathcal{L}_{ideal}(\theta) \leq \mathcal{L}_{\mathrm{DR}}(\theta) + |\mathrm{Bias}[\mathcal{L}_{\mathrm{DR}}(\theta)]| + \mathcal{B}(\mathcal{F})$$

$$\leq \mathcal{L}_{\mathrm{DR}}(\theta) + |\mathrm{Bias}[\mathcal{L}_{\mathrm{DR}}(\theta)]| + \left( 1 + \frac{1}{K_\psi} \right) \left( 2\mathcal{R}(\mathcal{F}) + K_\phi \sqrt{\frac{18\log(4/\eta)}{|\mathcal{D}|}} \right).$$

In Theorem 1, we have already prove that

$$|\mathrm{Bias}[\mathcal{E}_{\mathrm{DR}}(\theta)]| = \left| \frac{1}{|\mathcal{D}|} \sum_{(u,i) \in \mathcal{D}} \mathrm{Cov} \left( \frac{\hat{p}_{u,i} - o_{u,i}}{\hat{p}_{u,i}}, e_{u,i} - \hat{e}_{u,i} \right) \right.$$

$$\left. + \frac{1}{|\mathcal{D}|} \sum_{(u,i) \in \mathcal{D}} \left[ \left\{ 1 - \mathbb{E} \left[ \frac{o_{u,i}}{\hat{p}_{u,i}} \Big| x_{u,i} \right] \right\} \cdot \{ \mathbb{E}[e_{u,i} \mid x_{u,i}] - \mathbb{E}[\hat{e}_{u,i} \mid x_{u,i}] \} \right] \right|$$

$$\leq \left| \frac{1}{|\mathcal{D}|} \sum_{(u,i) \in \mathcal{D}} \mathrm{Cov} \left( \frac{o_{u,i} - \hat{p}_{u,i}}{\hat{p}_{u,i}}, e_{u,i} - \hat{e}_{u,i} \right) \right|$$

$$+ \frac{1}{|\mathcal{D}|} \sum_{(u,i) \in \mathcal{D}} \left| 1 - \mathbb{E} \left[ \frac{o_{u,i}}{\hat{p}_{u,i}} \Big| x_{u,i} \right] \right| \cdot \left| \mathbb{E}[e_{u,i} \mid x_{u,i}] - \mathbb{E}[\hat{e}_{u,i} \mid x_{u,i}] \right|,$$

therefore with probability at least $1 - \eta$, we have

$$\mathcal{L}_{ideal}(\theta) \leq \mathcal{L}_{\mathrm{DR}}(\theta) + \frac{1}{|\mathcal{D}|} \sum_{(u,i) \in \mathcal{D}} \left| 1 - \mathbb{E} \left[ \frac{o_{u,i}}{\hat{p}_{u,i}} \Big| x_{u,i} \right] \right| \cdot \left| \mathbb{E}[e_{u,i} \mid x_{u,i}] - \mathbb{E}[\hat{e}_{u,i} \mid x_{u,i}] \right|$$

$$+ \left| \frac{1}{|\mathcal{D}|} \sum_{(u,i) \in \mathcal{D}} \mathrm{Cov} \left( \frac{o_{u,i} - \hat{p}_{u,i}}{\hat{p}_{u,i}}, e_{u,i} - \hat{e}_{u,i} \right) \right| + \left( 1 + \frac{1}{K_\psi} \right) \left( 2\mathcal{R}(\mathcal{F}) + K_\phi \sqrt{\frac{18\log(4/\eta)}{|\mathcal{D}|}} \right),$$

which yields the stated results. $\qquad \square$

## B    MORE EXPERIMENT RESULTS

We change the sample ratios on **Yahoo** dataset to control the degree of overlap between users and items in the training and test set. The results are shown in Figure 5. The proposed SuperDR method still outperform baselines and achieve the promising debiasing performance.

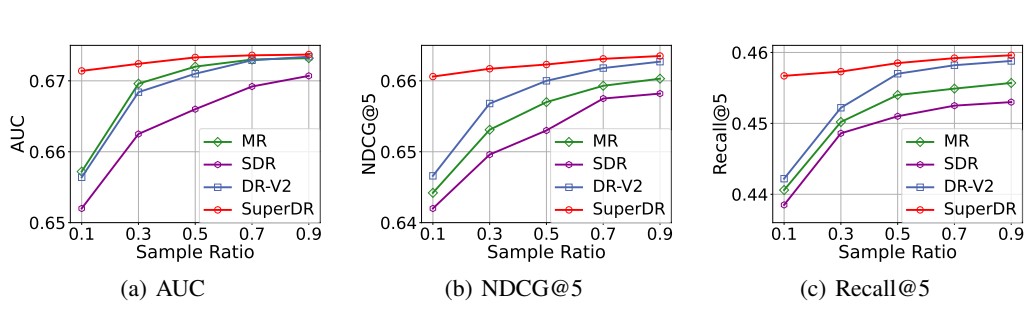

(a) AUC        (b) NDCG@5        (c) Recall@5

Figure 5: Effects of varying sample ratios on performance on the **Yahoo** dataset.

