# OpenReview forum: "Sampling Process Brings Additional Bias for Debiased Recommendation"
_ICLR.cc/2025/Conference — Submitted to ICLR 2025_

### Official Review · Reviewer_YHJN · 2024-11-01

**Soundness:** 3
**Presentation:** 3
**Contribution:** 3
**Rating:** 5
**Confidence:** 3

**Summary:**

This paper introduces a new doubly robust estimator to tackle the challenges when the users and the items are sampled from superpopulation. The designed imputation is weighted by the propensity score. Extensive experiments have been conducted in three datasets with various evaluation metrics.

**Strengths:**

1. The paper's presentation is well-structured and clear, making it easy for readers to follow the logical flow.
2. The authors provide a comprehensive theoretical analysis, rigorously demonstrating the advantages of their proposed methods over DR. The mathematical foundations are thoroughly explained and validated.
3. The experimental evaluation is robust, featuring diverse state-of-the-art baselines. The proposed approach achieves superior performance compared to existing methods, establishing new benchmarks in the field.

**Weaknesses:**

1. The distribution problem of the sampling lacks practical significance. Because the debiasing methods also trained on the full dataset, $D$, rather than only training on observed data.
2. The improvement of some metrics is quite low and is not statistically significant.
3. Some definitions of symbols are missed, for example, $m$ used in line 210.

**Questions:**

1. Can you compare relative debiasing work [1], including the performance and technical contribution?
2. Why are the performance improvements minor?
3. What are the details of the imputation model?

Ref:
[1] Ha, Mingming, et al. "Fine-Grained Dynamic Framework for Bias-Variance Joint Optimization on Data Missing Not at Random." arXiv preprint arXiv:2405.15403 (2024).

---

### Official Review · Reviewer_Nd1J · 2024-11-01

**Soundness:** 2
**Presentation:** 3
**Contribution:** 2
**Rating:** 3
**Confidence:** 3

**Summary:**

The paper addresses the issue of selection bias in recommender systems. The authors present a new approach to tackle biases caused by the sampling process, particularly in situations where users and items in training data do not match exactly with those in test data. This misalignment introduces additional biases in traditional doubly robust (DR) models. The authors propose a superpopulation-based DR model, which introduces mechanisms to control the biases introduced by sampling by modifying the imputation model to reduce generalization error. Extensive experiments on multiple real-world datasets demonstrate that SuperDR achieves more accurate and unbiased recommendations than existing DR approaches.

**Strengths:**

1. The paper presents a novel theoretical extension of the DR estimator, introducing a covariance control mechanism that provides unbiased recommendations even in scenarios with sampling-induced biases. This significantly advances the field of unbiased learning in recommendation systems, e.g., cold start users, cold start items.

2. The authors conduct thorough evaluations using three real-world datasets, including large-scale industrial data, which effectively show SuperDR’s advantage over conventional DR and other baselines.

**Weaknesses:**

1. In theorem 3, the SuperDR approach’s generalization upper bound is kind of loose. What is the $K_{\phi}$? It's better to clarify the definition and interpretation of $K_{\phi}$.

2. What is the Rademacher complexity for this model. Provide a more detailed analysis of the Rademacher complexity for this specific model. What is the insight from this bound? Without a tight bound, practitioners may not be able to fully trust the model’s debiasing performance across various scenarios, limiting the method’s utility for diverse recommendation tasks where model generalization is critical.
Discuss the practical implications of this bound for different recommendation scenarios. Compare this bound to bounds for other debiasing methods

3. Although large-scale datasets are used, the paper lacks a detailed discussion on the computational overhead and scalability of the proposed model in comparison to traditional DR models, which could impact its adoption.  Authors can provide the runtime comparisons between SuperDR and traditional DR models on datasets of varying sizes or memory usage analysis or discuss of how the method scales with increasing numbers of users/items

**Questions:**

1. In line 155, the unbiased definition is incorrect. The $\hat{e}$ is an estimator, but $e$ is a scalar, which is not comparable. Check the biased definition.

2. In line 210, what is the definition of m? How to get this m, by the EIB?

3. In line 214, what is v, a hyperparameter? What is $\phi$'s definition? Clarify v how it is set or tuned.
Provide a clear definition of  and explain its role in the model. This would improve the reproducibility of their work and help readers better understand the model's structure.

4. In line 419, how to understand even the sampling ratio is 1, there is still some benefit when you introducing some regularization?

---

### Official Review · Reviewer_BFu1 · 2024-11-03

**Soundness:** 3
**Presentation:** 2
**Contribution:** 3
**Rating:** 8
**Confidence:** 4

**Summary:**

This paper introduces Superpopulation Doubly Robust Target Learning (SuperDR), a method designed to address selection bias in recommendation systems when training and test data are sampled from a larger superpopulation with limited overlap. SuperDR extends traditional Doubly Robust (DR) methods by incorporating probabilistic imputation and propensity models to better handle uncertainty and variability across superpopulation samples. A key innovation in this approach is the inclusion of a covariance control mechanism within the loss function, which aims to reduce bias by minimizing the empirical covariance between imputation and propensity errors. Theoretical analysis suggests that controlling this covariance can help mitigate bias under superpopulation conditions, and the authors provide a generalization error bound to support this approach. Experiments on several recommendation datasets demonstrate that SuperDR outperforms traditional DR methods and other baselines, particularly in settings where train and test sets have limited overlap in user-item pairs.

**Strengths:**

__Innovative perspective for the source of bias in DR estimation__:
This paper pioneers the extension of Doubly Robust (DR) estimation to settings where training and test data are sampled from a superpopulation, meaning that there may be limited or no overlap in user-item pairs across the datasets. SuperDR specifically addresses scenarios where the training and test sets differ, which is common in real-world recommendation systems as user bases and item inventories evolve.

__Covariance Control to Address Bias from Imputation and Propensity Correlations__:
A major technical contribution is the introduction of a covariance control term within the loss function, which is designed to minimize empirical covariance between errors in propensity score estimation and imputed errors. In superpopulation contexts, these two types of errors can correlate, introducing additional bias in the DR estimator. By actively controlling this covariance, SuperDR reduces the bias resulting from these correlations, making the estimator more robust in settings with non-overlapping samples.

__Solid theoretical and Empirical Support for the Proposed Adjustments__:
The paper offers strong theoretical validation by deriving a generalization error bound using concentration inequalities. This bound supports SuperDR’s robustness claims under superpopulation sampling. Empirically, SuperDR is tested across several recommendation datasets and demonstrates improvements over traditional DR methods and other baselines. The ablation studies prove the effectiveness of the variance reduction techniques proposed.

**Weaknesses:**

__Some claims in the paper needs more clarifications__:
The paper claims " if the users and items in the training set are not exactly the same as those in the test set, all previous doubly robust based debiasing methods are biased". The paper’s statement oversimplifies the requirements for unbiasedness in DR estimation. In fact, DR methods do not require an _exact_ match between the users and items in the training and test sets. It might ask for positivity and overlap of supports [1] although an exact match surely satisfy this requirement.

The paper claims the added correction has no harm, while it is theoretically correct in the case of the empirically covariance drops to zero. However, in practical scenarios, where exact zero empirical covariance is challenging to achieve due to model imperfections, 𝜖 may not fully converge to zero and could introduce slight adjustments, potentially affecting the "no harm" property.

__The presentation of this paper can be improved__:
The lack of indexing in key formulas, especially when dealing with complex constructs like superpopulation, covariance terms, and probabilistic models, makes the paper harder to follow.

The paper has solid theoretical foundations, but it could benefit from more intuition and clearer explanations on why certain choices (like probabilistic models and covariance control) are necessary. The presentation would be stronger if it provided a more intuitive context for readers to understand how the superpopulation setting is motivated.




[1] Off-policy Bandits with Deficient Support, Noveen Sachdeva, Yi Su, Thorsten Joachims, KDD 20

**Questions:**

Q1: In Figure 2, the performance of SuperDR appears relatively stable even as the sample ratio decreases to 10%. What would happen if this ratio were further reduced to 5% or lower? While the model effectively reduces covariance introduced by the sampling process, a smaller dataset could lead to increased variance in both the imputation and propensity models, potentially impacting results negatively. However, this effect is not apparent in Figure 2. Could you elaborate on this?

Q2: Figure 4 demonstrates the reduction of covariance. Could you provide more details on how the covariance behaves for each individual dataset, and how it influences the model's overall performance? (also when you adjust it by sampling the data)

---

### Official Review · Reviewer_3Qqf · 2024-11-04

**Soundness:** 2
**Presentation:** 2
**Contribution:** 2
**Rating:** 3
**Confidence:** 4

**Summary:**

This paper points out that existing debiased recommendation methods usually assume that training and testing sets contain the same set of users and items, which may have limitations. It proposes a novel superpopulation doubly robust target learning method (SuperDR). Its core idea is to develop an additional pre-optimization process to control the covariance based on the bias terms of the doubly robust estimator derived under the superpopulation. The effectiveness of SuperDR is verified on three public datasets.

**Strengths:**

S1: Further improving the performance of debiased recommendations is valuable to enhance the trustworthiness of current recommender systems.

S2: Some theoretical insights into this research question are provided.

S3: Experimental results in various aspects are provided on multiple public datasets (including a dataset for industrial scenarios).

**Weaknesses:**

W1: The writing of the paper seems somewhat hasty, with some key details missing and some descriptions challenging to understand. Here are some examples:
* In Section 1, critical undescribed content is why existing debiased recommendation methods must be unbiased in a train-test setting with the same set of users and items. Based on the current description, it is not intuitive.

* For Section 4.1, why, in a general scenario, does the learned imputed error or propensity estimate become the expectation? It is also not intuitive. In Section 2, the authors also mentioned that this work extends the previous methods for deterministic error imputation and propensity models to probabilistic ones. This is hard to understand.

* In Section 4.2, the imputation balance correction calculation process is unclear. For example, what does $m$ mean? Is $\epsilon$ a value? What are the meanings of the two terms of this imputation balance correction? Why does it achieve the desired control process?

* What is the meaning of the superscript Bal used in many losses?

* No description is provided for the division of the data set in the experiment.

* In Section 5.1, MF generates embeddings for each user and item, which are then used as their features in MLP. The rationality of this process is questioned.

W2: The motivation proposed may not be convincing enough. In real-world scenarios, it is rare to directly use a model trained on user (or item) set A to predict user (or item) set B. This is inherently risky because the embeddings of external users (or items) are completely random and unlearned. In fact, if SuperDR does not use a pre-trained recommendation model (such as MF) to obtain a set of well-trained user and item embeddings in the experiment, I think it will also encounter this problem.

W3: The core contribution of the proposed solution compared to existing work is introducing an optimization process that controls covariance. However, there is a lack of more in-depth analysis, and the overall contribution is not significant enough.
* In Section 4.1, the bias term consists of two parts. Although the first part (i.e., covariance) is repeatedly emphasized, it is unclear how the second part is optimized in SuperDR.

* From an intuitive point of view, why does controlling for covariance help correct estimates of external populations? What is the connection between them?

* Based on Algorithm 1, SuperDR does not make many major changes to the DR architecture. This makes people worry that it may not have much gain compared to DR in some cases. In fact, based on the results in Table 1, we can find that its gain is indeed marginal, especially on some larger datasets. In addition, the pre-optimization process for covariance seems difficult, which may cause unstable training in some scenarios.

W4: The current experiments are not sufficiently powerful and have many limitations.
* As described in the previous concern, the process of generating features for users and items is somewhat unusual.

* In real scenarios, there is much feature information besides user and item IDs. It is unknown how they can be applied to SuperDR.

* Considering that current recommendation systems contain many representative architectures, using only a simple MLP as a skeleton model is flawed and lacks applicability.

* Debiased recommendation is a hot research topic with many recent works. Some newer methods (e.g. published in 2024) should be included as baselines.

**Questions:**

Please see the description in Weaknesses.

---

### Official Review · Reviewer_DAwb · 2024-11-04

**Soundness:** 2
**Presentation:** 2
**Contribution:** 2
**Rating:** 3
**Confidence:** 3

**Summary:**

The authors first derive the bias of existing DR methods under a "superpopulation" scenario, where users and items differ between training and test sets.

**Strengths:**

1.	The paper is well written and easy to follow.
2.	The paper provides theoretical proof and experimental results of the SuperDR, demonstrating its robustness and effectiveness across different datasets.

**Weaknesses:**

1.	The most critical issue is that this paper assumes the users or items in the training set and test set are different. I don’t quite understand the rationale behind this assumption. Why does selection bias exist for newly added users/items that have not yet been interacted with?
2.	The appearance of new users/items in the test set is essentially a cold-start problem. Could the authors compare their results with existing cold-start methods? Alternatively, could this approach also be plugged into cold-start methods?
3.	Why is the dataset size so small? Why not use a larger dataset, such as the Amazon dataset or others?
4.	In the experiment, when the sampling rate is not equal to 1, how are the embeddings for user/item that have not appeared in the training set handled? Are they initialized completely randomly?

**Questions:**

See Weakness.

---

### Meta-Review · Area_Chair_o37Y · 2024-12-18

**Metareview:**

The submission initially received negative feedback from reviewers, and the authors did not engage in the discussion of the rebuttal phase. The current version can't be accepted as an ICLR paper.

I have summarized the primary issues that need to be addressed:

1. **Validation of Assumptions**: The authors make several assumptions regarding the training and testing data that require validation.

2. **Writing and Clarity**: The manuscript needs significant improvement in clarity and coherence. Key technical details and contributions should be clearly articulated to enhance the reader's understanding.

3. **Rademacher Complexity**: The paper presents a loose bound, and the upper bound for Rademacher complexity is absent.

4. **Experimental Significance**: The significance of the experiments conducted is relatively low, primarily due to the use of somewhat small datasets. Expanding the dataset or enhancing the experimental design could provide more robust findings.

Based on these concerns, I recommend rejecting the submission. However, I encourage the authors to consider the reviewers' suggestions, as they provide valuable insights to refine and strengthen their work for future submissions.

**Additional Comments On Reviewer Discussion:**

The authors didn't participate in the rebuttal.  There are many issues in the current version, I have to recommend rejecting it.

---

### Decision · Program_Chairs · 2025-01-22

Reject